# First Pan-Arctic Assessment of Dissolved Organic Carbon in Lakes of the Permafrost Region

Lydia Stolpmann[1,2], Caroline Coch[1,2,3], Anne Morgenstern[1], Julia Boike[1,4], Michael Fritz[1], Ulrike Herzschuh[1,2,5], Kathleen Stoof-Leichsenring[1], Yury Dvornikov[6], Birgit Heim[1], Josefine Lenz[1,7], Amy Larsen[8], Katey Walter Anthony[7], Benjamin Jones[9], Karen Frey[10], and Guido Grosse[1,2]

[1]Alfred Wegener Institute, Helmholtz Centre for Polar and Marine Research, Potsdam, Germany
[2]Institute of Geosciences, University of Potsdam, Potsdam, Germany
[3]World Wildlife Fund, The Living Planet Centre, Rufford House, Brewery Road, Working, Surrey, GU214LL
[4]Geography Department, Humboldt-Universität zu Berlin, Germany
[5]Institute of Biochemistry and Biology, University of Potsdam, Potsdam, Germany
[6]Agrarian-Technological Institute, Peoples' Friendship University of Russia, Moscow, 117198, Russia
[7]Water and Environmental Research Center, University of Alaska, Fairbanks, AK, USA 99775
[8]Yukon-Charley Rivers National Preserve and Gates of the Arctic National Park and Preserve, National Park Service, Fairbanks, AK, USA
[9]Institute of Northern Engineering, University of Alaska, Fairbanks, AK, USA
[10]Graduate School of Geography, Clark University, Worcester, MA, USA

*Correspondence to*: Lydia Stolpmann (Lydia.Stolpmann@awi.de)

**Abstract.** Lakes in permafrost regions are dynamic landscape components and play an important role for climate change feedbacks. Lake processes such as mineralization and flocculation of dissolved organic carbon (DOC), one of the main carbon fractions in lakes, contribute to the greenhouse effect and are part of the global carbon cycle. These processes are in focus of climate research but studies so far are limited to specific study regions. In our synthesis, we analysed 2,167 water samples from 1,833 lakes across the Arctic in permafrost regions of Alaska, Canada, Greenland, and Siberia to provide first pan-Arctic insights for linkages between DOC concentrations and the environment. Using published data and unpublished datasets from the author team we report regional DOC differences linked to latitude, permafrost zones, ecoregions, geology, near-surface soil organic carbon contents, and ground ice classification of each lake region. The lake DOC concentrations in our dataset range from 0 mg L$^{-1}$ to 1,130 mg L$^{-1}$ (10.8 mg L$^{-1}$ median DOC concentration). Regarding the permafrost regions of our synthesis, we found median lake DOC concentrations of 12.4 mg L$^{-1}$ (Siberia), 12.3 mg L$^{-1}$ (Alaska), 10.3 mg-L$^{-1}$ (Greenland), and 4.5 mg L$^{-1}$ (Canada). Our synthesis shows a significant relationship between lake DOC concentration and lake ecoregion. We found higher lake DOC concentrations in boreal permafrost sites compared to tundra sites. We found significantly higher DOC concentrations in lakes in regions with ice-rich syngenetic permafrost deposits (yedoma) compared to non-yedoma lakes and a weak significant relationship between soil organic carbon content and lake DOC concentration as well as between ground ice content and lake DOC. Our pan-Arctic dataset shows that the DOC concentration of a lake depends on its environmental properties, especially on permafrost extent and ecoregion, as well as vegetation, which is the most important driver of lake DOC in this study. This new dataset will be fundamental to quantify a pan-Arctic lake DOC pool for estimations of the impact of lake DOC on the global carbon cycle and climate change.

# 1 Introduction

In northern high latitudes, where mean annual ground temperatures are below 0 °C, permafrost has been an important carbon (C) sink for thousands of years since freezing is one of the most effective mechanisms for long-term C fixation in soils (Schuur et al., 2008; Grosse et al., 2011). Permafrost landscapes store large amounts (~1,300 to 1,600 Pg C) of soil organic C (Hugelius et al., 2014) and are a potential source for C emissions to the atmosphere when soil temperatures exceed 0 °C and permafrost thaws (McGuire et al., 2009; Koven et al., 2011). Through recent climate change, Arctic permafrost regions experienced an increase of permafrost temperatures by 0.5 to 2 °C and a local deepening of the active layer of up to 90 cm since the 1970s (Romanovsky et al., 2010; IPCC, 2013; Biskaborn et al., 2019). More recently, permafrost warmed globally by an average of 0.29 °C +/- 0.12 °C over the 2007-2016 period due to higher air temperatures, with some of the strongest warming trends (about 0.9 °C per decade) measured in individual boreholes at the polar stations Marre Sale in northwest Siberia and Samoylov Island in northeast Siberia (Biskaborn et al., 2019). In addition, thermokarst and thermo-erosion processes act as a mechanism for the rapid release of permafrost C in the climate system (Walter Anthony et al., 2018; Turetsky et al., 2020). Hence, the impact of global climate change on permafrost regions and their C cycling has to be thoroughly investigated.

Of particular interest is ice-rich permafrost, which is vulnerable to rapid degradation processes, such as thermokarst and thermo-erosion that lead to ground ice melt, subsequent soil volume loss, and ground subsidence. Consequently, characteristic landforms such as thermo-erosional valleys, thaw slumps, and thermokarst lakes form in these regions. Thermokarst lakes are quite dynamic and widespread landscape features in the Arctic (Jones et al., 2011; Grosse et al., 2013; Manasypov et al., 2015), and their biochemical processes play an important role for C cycling and climate change feedbacks in the Arctic and beyond (Walter Anthony et al., 2018).

In lakes, dissolved organic carbon (DOC) is one of the main C fractions (Tranvik et al., 2009). It is mobile and can be chemically labile (Vonk et al., 2013a, b). DOC in lakes can be produced in the lake itself (autochthonuous DOC) or in the catchment of the lake (allochthonuous DOC) (Sobek et al., 2007). The organic carbon (OC) content of terrestrial soils is the main source for allochthonuous DOC. DOC in lakes can be transferred to and stored in lake sediments due to flocculation (Tranvik et al., 2009). DOC can also be degraded by photo oxidation or microbial activity, resulting in the mineralization of OC to carbon dioxide ($CO_2$) and methane ($CH_4$) and the emission to the atmosphere (Frey & Smith, 2005; Battin et al., 2008; Tranvik et al., 2009; Vonk et al., 2013a, b). These processes are important components of the northern C cycle and affect greenhouse gas emissions from lakes. Vonk et al. (2015) suggested that the C flux from surface waters to the atmosphere and from land to ocean represents roughly one third to one half of the net C exchange from land to the atmosphere in the Arctic. Numerous studies estimated OC pools in Arctic soils (Zimov et al., 2006; Strauss et al., 2013; Hugelius et al., 2014, Hugelius et al., 2020) while others investigated DOC and its release from northern high latitude soils and ground ice (Freeman et al., 2004; Wickland et al., 2007; Prokushkin et al., 2009; Fritz et al., 2015; Tanski et al., 2016).

In recent years, DOC concentrations, lability and mobility in arctic lake systems, including thermokarst lakes, have been investigated; however, these studies have largely been limited to specific regions. For example, it was found that hydrologic

linkages between a pond and its catchment affect the load of DOC in ponds in northern Siberia (Abnizova et al., 2014), and that DOC in different lake-basin types responds differently to climate change (Larsen et al., 2017). For specific regions of West-Siberia, Shirokova et al. (2013) found a negative correlation between DOC concentration and the size and age of thermokarst lakes. Among global lakes (7500 lakes from 35 land-cover types), Sobek et al. (2007) found no correlation between lake area or other lake properties and DOC concentration, but DOC concentration in lakes was found to depend on catchment properties such as topography and climate. However, permafrost-region lakes, which represent approximately 25 % of global lakes (Lehner & Döll, 2004), only comprised about 10 % of the 7,500 global lakes studied with respect to DOC (Sobek et al., 2007). Hence, a pan-Arctic focused analysis of the spatial variability of lake DOC in permafrost regions is still missing.

The objectives of this study are to synthesize existing datasets of lake DOC in northern permafrost regions, to provide first insights for linkages between DOC concentration and environmental parameters (permafrost zone, ecoregion, deposit types, ground ice content and soil organic carbon content), and to identify drivers for lake DOC concentration in this region affected by rapid climate change. Our synthesis includes published datasets as well as unpublished datasets from the author team to find regional differences in DOC concentration of lakes across the Arctic.

## 2 Study areas

In our synthesis, we included 2,167 samples from 1,833 lakes of 13 study areas (22 sites) across the Arctic, sampled from year 1979 to 2017 (Table 1, Fig. A1). Lakes in our study are located from 59.2° to 82.5° northern latitude. 49.3 % of our dataset come from sites in Alaska, 24.2 % from Canada, 23.3 % from Siberia and 3.2 % from Greenland. The study areas of our dataset are dominated by tundra climate and very cold subarctic climate. The Nunavut study area is also characterized by cool continental climate. The mean annual air temperature of our study areas ranges from -18 °C in the Canadian Arctic Archipelago (Michel, 2011) to -0.7 °C in Whitehorse, Yukon (Bonnaventure & Lewkowicz, 2011). All study lakes are located in landscapes influenced by permafrost (Fig. 1a). Lakes in this synthesis cover the full range of permafrost extents from continuous, discontinuous, isolated, and sporadic permafrost areas.

Study sites of the North and Northwest Alaska study area (Fig. 1a, 1-3) are predominantly located in the continuous permafrost zone (82 % of the lakes studied in this area). 46 % of the studied lakes in this study area are located in the tundra ecoregion and 54 % in the tundra-boreal transition region. The North and Northwest Alaska study area is mainly composed of fluvial and yedoma deposits (62 %). The Southcentral Alaska study area (Fig. 1a, 4) is predominantly underlain by discontinuous permafrost. Studied lakes in this study area are located in the boreal ecoregion and are surrounded by glacio-moraine (67 %) and mountain-alluvium (13 %) deposits. The study sites in the Interior Alaska study area (Fig. 1a, 5-6) are predominantly located in the discontinuous permafrost zone (65 %), 19 % of studied lakes in this study area are located in the isolated permafrost zone, belonging to the Denali National Park and Preserve (Fig. 1a, 5). The Interior Alaska study area is situated in the boreal zone and mainly underlain by fluvial (55 %) and yedoma (16 %) deposits.

The study sites in the Yukon and Northwest Territories study areas (Fig. 1a, 7-10) are predominantly situated in the continuous permafrost zone (65 % of studied lakes in this area), the discontinuous permafrost zone (30 %), and some lakes in the sporadic permafrost zone (5 %), located in the Whitehorse transect.

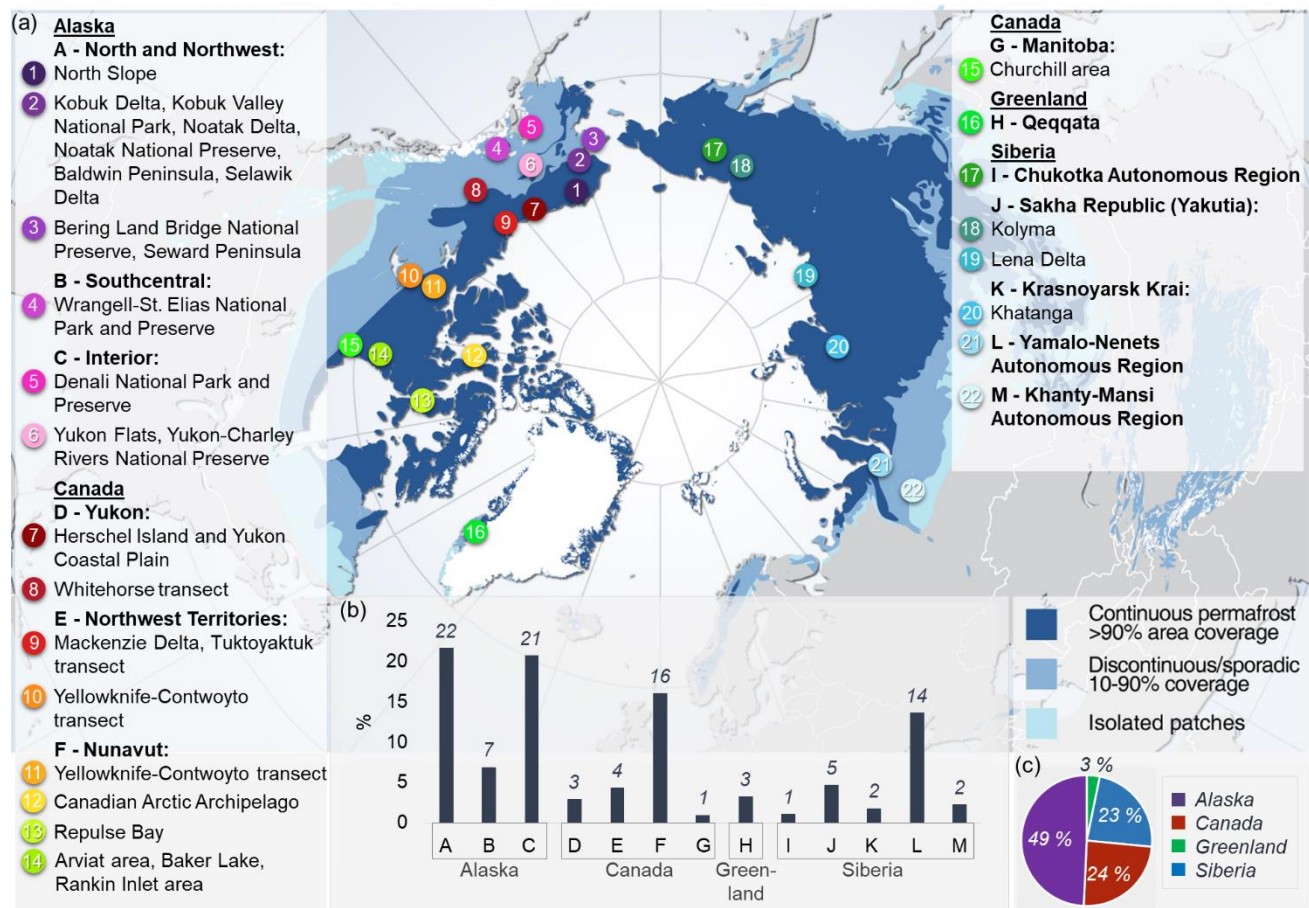

**Figure 1: Overview of study regions (underlined bold font), study areas (bold font) and sites overlain of the map of permafrost zones (a), histogram of the amount of lakes in percentage by the study area in our dataset (b), and pie chart of lake distribution in the dataset by overarching regions (c) (Background map: after Brown et al., 1997).**

10 Studied lakes of this study area can be found in the tundra ecoregion, in the boreal-tundra transition zone and in the boreal forest, with glacial deposits. Study sites in the Nunavut study area (Fig. 1a, 11-14) are located in the zone of continuous permafrost. Studied lakes in this area are situated in the tundra ecoregion and are surrounded by glacial, bedrock and colluvial deposits. Studied lakes of the Manitoba study area (Fig. 1a, 15) are located in the continuous permafrost zone and are predominantly situated in the boreal forest, underlain by glacio-marine deposits.

The Qeqqata study area (Fig. 1a, 16) in Greenland is situated in the continuous permafrost zone. Studied lakes in this area are located in the tundra ecoregion and surrounded by aeolian deposits.

The Siberian Yamalo-Nenets Autonomous Region (A.R.) study area (Fig. 1a, 21) covers the continuous, discontinuous and sporadic permafrost zones. Here, 72 % of the studied lakes are situated in boreal forest and 28 % in the tundra ecoregion, especially on the Yamal Peninsula. The Yamalo-Nenets A.R. study area is underlain by glacial-morain, glacio-lacustrine, glacio-fluvial and alluvial deposits. Studied lakes of the Khanty-Mansi A.R. study area (Fig. 1a, 22) are situated in the continuous and isolated permafrost zone. This area is situated in the boreal forest and dominated by glacio-fluvial deposits.

**Table 1: Overview of temporal sample distribution for each study area. Number in brackets are numbers of samples collected in each year.**

|  | Study area | Months of sample collection | Years of sample collection (number of samples) |
|---|---|---|---|
| Greenland | Qeqqata | April – August | 2002 (10), 2003 (23), 2009 (5), 2013(20), 2014(23) |
| Siberia | Yamalo-Nenets Autonomous Region | June - August | 1999 (8), 2000 (17), 2001 (1), 2010 (14), 2011 (43), 2013 (93), 2015 (24), 2016 (49) |
|  | Khanty-Mansi Autonomous Region | July - August | 1999 (21), 2000 (1), 2001 (1), 2016 (18) |
|  | Chukotka Autonomous Region | July | 2016 (20) |
|  | Krasnoyarsk Krai | July - August | 2013 (32) |
|  | Sakha Republic (Yakutia) | July - October | 2002 (27), 2013 (3), 2014 (38), 2016 (59) |
| Canada | Yukon | July - August | 1990 (22), 2012 (3), 2013 (6), 2014 (22), 2015 (1) |
|  | Northwest Territories | July, September | 1990 (37), 1991 (20), 2004 (22) |
|  | Nunavut | June - September | 1979 (5), 1980 (17), 1983 (17), 1984 (2), 1985 (21), 1989 (22), 1990 (6), 1991 (4), 1992 (17), 1993 (22), 1994 (22), 1995 (20), 1996 (5), 1997 (7), 2017 (19) |
|  | Manitoba | July - August | 2006-2010 (17) |
| Alaska | North and Northwest | June - September | 2008 (15), 2009 (30), 2010 (2), 2011 (71), 2012 (98), 2013 (56), 2014 (115), 2015 (10), 2016 (102) |
|  | Southcentral | May - September | 2009 (25), 2010 (9), 2011 (96), 2015 (4), 2016 (4) |
|  | Interior | May - September | 2003 (13), 2004 (14), 2005 (14), 2006 (30), 2007 (35), 2008 (92), 2010 (65), 2011 (63), 2012 (60), 2013 (36), 2014 (13), 2015 (20), 2016 (72) |

The Chukotka A.R. study area (Fig. 1a, 17) is situated in the continuous permafrost zone and covers the full range of tundra ecoregion, boreal forest and tundra-boreal transition region. Lakes in this study area are surrounded by ice-rich syngenetic permafrost deposits (yedoma) and fluvial deposits. The Khatanga study site in the Krasnoyarsk Krai study area (Fig. 1a, 20) is located in the continuous permafrost zone. This area is situated in the tundra ecoregion and underlain by lacustrine, alluvial and eluvial deposits. The Sakha Republic (Yakutia) study area (Fig. 1, 18-19) includes sites in the Lena River Delta (Kurungnakh Island, Sobo-Sise Island, Samoylov Island, and Bykovsky Peninsula) and sites close to the Kolyma River. These sites are situated in the continuous permafrost zone. The Lena Delta study site is situated in the tundra ecoregion, whereas the Kolyma study site is situated in the boreal forest. The study lakes of the Sakha Republic study area are mainly located in ice-rich syngenetic permafrost deposits (yedoma), or fluvial and alluvial deposits.

## 3 Methods

### 3.1 Data extraction from existing studies

For this synthesis, we searched the scientific literature for the keywords DOC and lakes in permafrost regions and largely focused on local to regional lake DOC syntheses that provided data at the individual lake level (i.e., not averaged values for groups of lakes or regions). From identified references (Pienitz et al., 1997a,b; Hamilton et al., 2001; Lim et al., 2001; Kokelj et al., 2005; Medeiros et al., 2012; Halm & Griffith, 2014; Manasypov et al., 2014; Manasypov et al., 2015; Northington & Saros, 2016; Larsen et al., 2017; Osburn et al., 2017; Coch et al., 2019; Serikova et al., 2019; Johnston et al., 2020) data for 1,757 DOC samples, collected from 1,478 individual lakes, were extracted into a database for further analysis. Unpublished field data of the author team was included in the database (410 samples from 355 lakes). The database includes samples that have been collected during the period of April to early October (Table 1). If there was a lake sampled once in a month for more than one year, we calculated the average lake DOC concentration. Samples from the author team were taken from or near the water surface as well as the vast majority of the synthesized data. Although, some of the synthesized data do not provide the sampling depth we can assume that the majority of these arctic lakes and ponds are shallow and well mixed. Across our synthesis dataset, various well-established methods (Bauer & Bianchi, 2011) were used to quantify DOC concentration, including high-temperature catalytic combustion, low-temperature chemical oxidation, and photochemical oxidation. The 246 samples from Alfred Wegener Institute (AWI), Helmholtz Centre for Polar and Marine Research, were analysed with high-temperature catalytic combustion, described in Appendix A.

### 3.2 Sample database and geospatial analysis

We created a geospatial database of permafrost-region lakes with DOC data (PeRL-DOCv1) in the desktop Geoinformation System (GIS) ArcMap (10.4.1, ESRI) containing all 1,833 lakes as point features. Additional data layers were included in the PeRL-DOCv1 GIS for the analysis of lake environmental characteristics, including layers on permafrost and ground ice

distribution (Jorgenson et al., 2008), surface geology (Jorgenson et al., 2008), and yedoma distribution (Strauss et al., 2016). For all lakes, a range of parameters (Table A1) was extracted and exported into the spreadsheet database for further analysis. For the determination of yedoma and non-yedoma areas, we used the Database of Ice-Rich Yedoma Permafrost (IRYP) by Strauss et al. (2016). By using the study site descriptions from the synthesized lake DOC literature and a map of terrestrial

ecoregions (Olson et al., 2004), we assigned an ecoregion for each data point.

For inferring lake genesis, each data point was assigned a deposit type, which refers to the surrounding deposit type of each lake. For this, we used the Permafrost characteristics of Alaska map by Jorgenson et al. (2008) for Alaska, Nielsen (2010) for Greenland, the Map of the Quaternary Formations of the Russian Federation (Petrov et al., 2014), the Geological Survey of Canada map of Fulton (1995) for Canada, and the yedoma distribution database of IRYP (Strauss et al., 2016). Furthermore,

we added the ice content for the surrounding area of each lake, using the term 'low', 'moderate', 'high' and 'variable' (Jorgenson et al., 2008; Brown et al., 1997). Finally, we used the Northern Circumpolar Soil Carbon Database (NCSCDv2) to add the soil organic carbon content (SOCC) of the area surrounding each lake for the upper 0 to 100 cm, 100 to 200 cm, 200 to 300 cm, and aggregated for the upper 300 cm of soil (Hugelius et al., 2014).

### 3.3 Statistical Analysis

To conduct statistical tests, we used RStudio (version 1.0.153). We tested normality by using the Shapiro–Wilk test. Because our data does not follow a normal distribution, we used the Spearman rank correlation coefficient ($\rho$) to measure each relationship between DOC concentration and a further parameter (latitude, permafrost zone, ecoregion, ground ice content, deposit type, SOCC) for all lakes in our dataset. We used the Wilcoxon-Mann-Whitney test to determine the difference in means between two populations. To analyse the relationship of DOC and multiple parameters we performed a principal

component analysis (PCA). Our dataset contains six samples from Qeqqata on Greenland (Osburn et al., 2017), collected in April with under-ice conditions. For the sake of comparability, these data have not been included in the statistical analysis.

## 4 Results

### 4.1 Temporal variability of DOC concentration data

For only 81 of 1,833 lakes in our dataset we had multi-temporal data, which means that these lakes were sampled at least two

25  times during the ice-free period.

For 42 % of the multi-temporal subset we found increasing DOC concentrations in a year, regarding the variation of sub annual samples. For 42 % of the multi-temporal subset we found decreasing DOC concentrations and for 6 % of the multi-temporal subset we found fluctuating values in sub-annual samples. In some cases, the DOC concentration increased after snowmelt and further decreased until fall or decreased in summer and increased until fall.

In our dataset, 16 lakes were sampled multiple times over the same seasonal period in the study site North Slope in North Alaska and six lakes were sampled multiple times over the same seasonal period in the study area Qeqqata, Greenland (Osburn

et al., 2017). The six lakes located in Qeqqata were sampled in April, June and August in 2014, whereas lakes on North Slope were sampled in mid-June, end-June, in July and August in 2014. For five of the six lakes in Qeqqata, the highest DOC concentration of the respective sampling series was found for April samples. Then, the DOC concentration decreased in June and increased in August (Table 2). For these lakes, a 30 % to 45 % higher DOC concentration in April and up to 25 % higher DOC concentration in August was observed in comparison to the June sampling and therefore demonstrates a seasonal DOC variability.

Table 2: DOC concentration of six lakes from Qeqqata, Greenland, sampled three times in 2014 (Osburn et al., 2017) and 16 lakes from North Slope, Alaska, samples four times in 2014.

| Region | Study area | Lake name | DOC concentration [mg L$^{-1}$] | | | | |
|---|---|---|---|---|---|---|---|
| | | | April | Mid-June | End-June | July | August |
| Greenland | Qeqqata | SS906 | 8 | 5.2 | | | 5.8 |
| | | SS1381 | 39.2 | 24 | | | 31.3 |
| | | SS2 | 35.1 | 23.8 | | | 27.7 |
| | | SS8 | 52.5 | 28.7 | | | 38.3 |
| | | SS904 | 8.1 | 5.2 | | | 5.8 |
| | | SS1590 | 31.1 | 36.6 | | | 25 |
| Alaska | North and Northwest | Hannahbear | | 5.5 | 6.5 | 2.7 | 1.7 |
| | | Ini-001 | | 0.4 | 2.5 | 0 | 0.2 |
| | | Ini-002 | | 12.9 | 20.8 | 19.7 | 16.7 |
| | | Ini-003 | | 0.5 | 0.3 | 0 | 0 |
| | | Ini-004 | | 6.5 | 4.5 | 0.7 | 0.4 |
| | | Ini-005 | | 4.7 | 4.7 | 3.4 | 3.1 |
| | | Ini-006 | | 0 | 0.2 | 0 | 0 |
| | | LonelyWolf | | 1.6 | 2.1 | 0.1 | 1.3 |
| | | CrazyBear | | 0.9 | 2.2 | 1.6 | 0 |
| | | Duckfish | | 2.1 | 1.4 | 0.7 | 0.7 |
| | | FC-L9811 | | 0 | 1.9 | 0 | |
| | | FC-L9819 | | 3.6 | 1.2 | 0 | |
| | | FC-L9820 | | 4.6 | 6.5 | 2.7 | 2.5 |
| | | FC-M9925 | | 2.5 | 2.6 | 1.3 | 2.3 |
| | | FC-MC7916 | | 5.3 | 3.5 | 0.8 | 0.6 |
| | | FC-R0066 | | 0.8 | 1.9 | 8.7 | 0 |

In contrast to the Qeqqata samples we found decreasing DOC concentrations in 12 of 16 lakes on the North Slope comparing DOC concentrations of mid-June and August samples (Table 2). We also checked for seasonal variability in a larger dataset available from the study areas Southcentral and Interior Alaska where different sets of lakes were sampled during each month from May to September. This allowed an analysis of the median DOC concentration for each month for each of the two study areas. For Southcentral Alaska we found a pattern similar to that in Qeqqata with a 17 % higher DOC concentration in May and September compared to July (Table A2). Additionally, we compared samples of the whole dataset from the months June and August. For these months, in addition to the Qeqqata and North Slope samples, samples from the study areas Yamalo-Nenets A.R., North and Northwest Alaska, Southcentral and Interior Alaska were available. In three of the four study areas we also found higher DOC concentrations in August than in June, comparable to the Qeqqata lakes.

## 4.2 Variable DOC concentrations across the Arctic

Lakes in our database from sites across the Arctic, covering different permafrost zones, ecoregions and deposit types, show a high variation of lake DOC concentration. We found differences between the four regions of Alaska, Canada, Greenland and Siberia, as well as between study areas and study sites within these regions (Fig. 2, Fig. 3, Table A3). The median DOC concentration across the entire dataset was 10.8 mg L$^{-1}$. The concentration ranged from 0 mg L$^{-1}$ to 1,130 mg L$^{-1}$ (Table 3). 91.8 % of the lakes included in our dataset have a DOC concentration between 0 and 30 mg L$^{-1}$. Comparing DOC concentrations of lake water in permafrost regions of Alaska, Canada, Greenland and Siberia, we found median DOC concentrations of 12.3 mg L$^{-1}$, 4.2 mg L$^{-1}$, 10.3 mg L$^{-1}$ and 12.4 mg L$^{-1}$, respectively.

**Table 3: DOC concentrations according to study sites.**

|  | Study area | No. of samples/ No. of lakes | DOC concentration [mg L$^{-1}$] | | |
|---|---|---|---|---|---|
|  |  |  | range | mean | median |
| Greenland | Qeqqata | 81/59 | 1-61.3 | 18.5 | 10.3 |
| Siberia | Yamalo-Nenets A.R. | 249/249 | 3.2-63.4 | 18.1 | 15.6 |
|  | Khanty-Mansi A.R. | 41/41 | 5.8-36.1 | 13.7 | 11 |
|  | Chukotka A.R. | 20/20 | 1.1-19.6 | 9.5 | 9.6 |
|  | Krasnoyarsk Krai | 32/32 | 2.3-19.4 | 8.3 | 8.3 |
|  | Sakha Republic (Yakutia) | 127/85 | 2.4-33.3 | 9.6 | 9.8 |
| Canada | Yukon | 54/54 | 3.1-38.7 | 15.4 | 14.7 |
|  | Northwest Territories | 79/79 | 1.7-30 | 10.2 | 9.1 |
|  | Nunavut | 302/294 | 0-31.9 | 3.9 | 2.9 |
|  | Manitoba | 17/17 | 2.7-21.2 | 9.1 | 7 |
| Alaska | North and Northwest | 499/397 | 0-53.3 | 9.6 | 8.6 |
|  | Southcentral | 138/126 | 0.8-36.8 | 14.1 | 13.8 |
|  | Interior | 528/380 | 1.4-1,130 | 25 | 16.8 |
|  | Total | 2,167/1,833 | 0-1,130 | 14.3 | 10.8 |

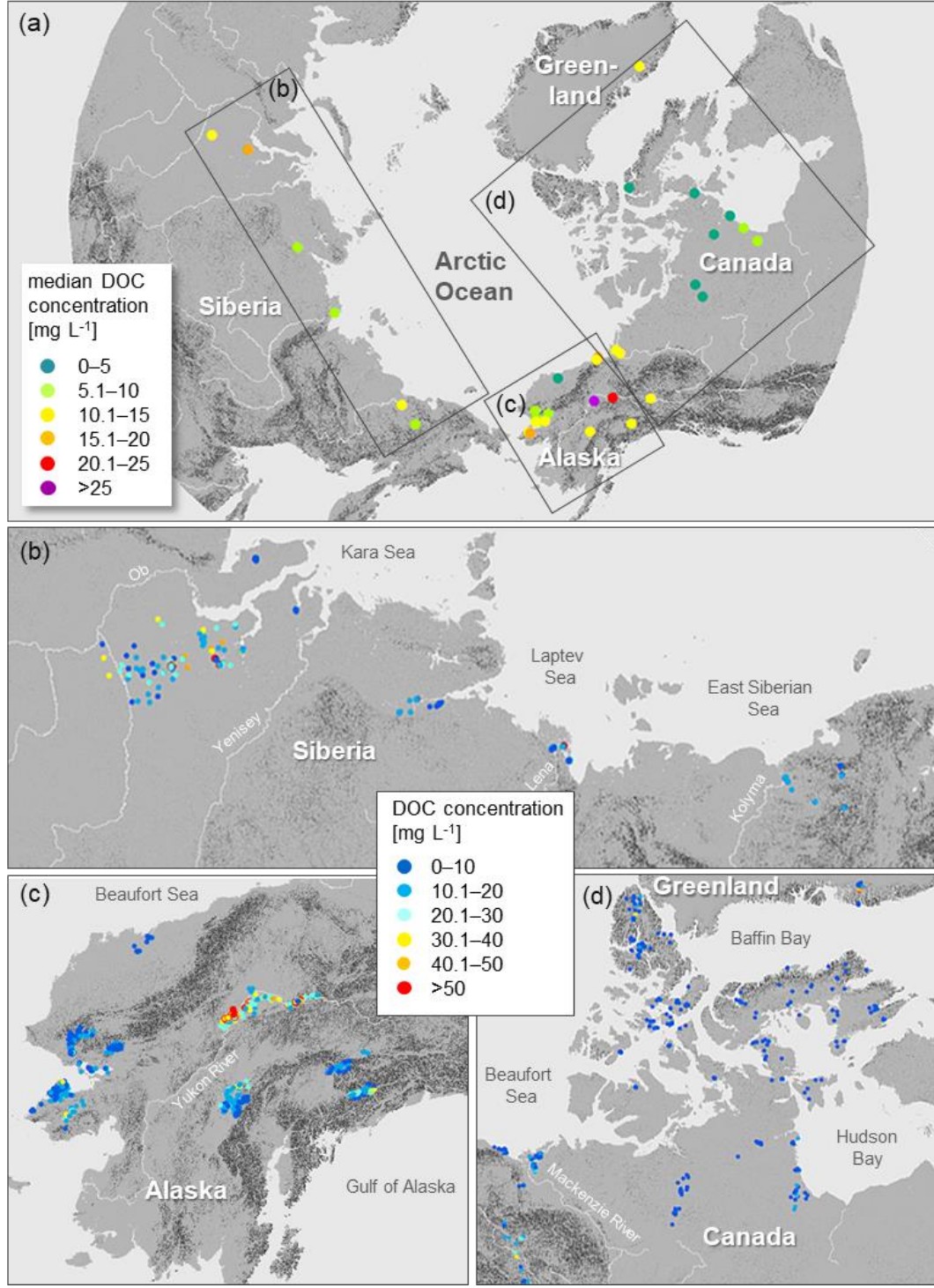

**Figure 2: Map of lake DOC concentrations (mg L⁻¹) and regional variability. Median DOC concentration for each study site (a). DOC concentrations of individual lakes in the study regions Siberia (b), Alaska (c) and Canada and Greenland (d) (Background map: ESRI).**

Figure 3 highlights the variability of median DOC concentration in the permafrost regions of Alaska, Canada, Greenland and Siberia, and demonstrates the large range of DOC concentration in Alaska. In contrast, lakes in the Canadian permafrost region had a smaller range of DOC concentrations (Fig. 2d). We found that 80.3 % of samples collected in Canadian lakes had a lower DOC concentration than the dataset median of 10.8 mg $L^{-1}$. In Alaska and Siberia, we found that about 58 % of the lakes

had higher DOC concentrations than the dataset median. Lakes in Greenland showed a 50:50 ratio with DOC concentrations below and above the dataset median. A large number of lakes with DOC concentrations above 30 mg $L^{-1}$ were found in Interior Alaska in the Yukon Flats and Yukon-Charley Rivers National Preserve (Fig. 2c). We had four lakes with strikingly high DOC concentrations more than ten times higher than the dataset median. These concentrations are 1,130 mg $L^{-1}$, 507 mg $L^{-1}$, 433 mg $L^{-1}$ and 173 mg $L^{-1}$ and all four lakes were located in the Yukon Flats in Interior Alaska. In addition, about 25 % of

lakes with a DOC concentration above 30 mg $L^{-1}$ were located in the Yamalo-Nenets A.R. (Fig. 2b). We found that lake DOC concentration was negatively correlated with geographic latitude of a lake ($\rho = -0.3$; $p < 0.05$; Table 4; Fig. A2). The DOC concentration of lakes in the southernmost study sites (Yukon Flats and Yukon-Charley Rivers National Preserve) showed a large range from 10.2 to 1,300 mg $L^{-1}$, and 5.0 to 66.7 mg $L^{-1}$, respectively (Table A3).

**Table 4: Results of the Spearman Rank correlation, testing the relationship between lake DOC concentration and lake parameters.**

|  | latitude | permafrost zone | ecoregion | ground ice content | deposit type | SOCC 0-300 cm | SOCC 0-100 cm |
|---|---|---|---|---|---|---|---|
| $\rho$ | -0.3 | 0.37 | 0.31 | 0.05 | -0.2 | 0.09 | 0.12 |
| $p$ | < 0.05 | < 0.05 | < 0.05 | < 0.05 | < 0.05 | < 0.05 | < 0.05 |

**4.3 Higher DOC concentrations in boreal forest lakes**

In our dataset, 43.7 % of the lakes were located in the boreal forest ecoregion, 42.6 % in the tundra region, and 13.7 % in a boreal-tundra transition zone. We found a significant relationship between lake DOC concentration and the lake surrounding

ecoregion ($\rho = 0.31$; $p < 0.05$; Table 4; Fig. A2), with significantly lower DOC concentrations in lakes of the tundra region ($p < 0.05$). The DOC concentration of lakes in the boreal zone ranged from 0.8 mg $L^{-1}$ to 1,130 mg $L^{-1}$ and the median DOC concentration in the boreal zone was 15.3 mg $L^{-1}$, whereas the DOC concentration of lakes in the tundra zone ranged from 0 mg $L^{-1}$ to 816 mg $L^{-1}$ with a median of 6.8 mg $L^{-1}$ (Fig. 3). With a median DOC concentration of 8.5 mg $L^{-1}$, lakes in the boreal-tundra transition zone had significantly lower DOC concentrations than lakes in the boreal forest ($p < 0.05$).

**4.4 Lower DOC concentrations in lakes of the continuous permafrost zone**

Median DOC concentration was highest in lakes of the sporadic permafrost zone (17.3 mg L$^{-1}$) and negatively correlated with permafrost extent ($\rho = 0.37$; $p < 0.05$; Fig. 3; Table 4; Fig. A2). DOC concentrations in lakes of the discontinuous zone were significantly higher (14 mg L$^{-1}$) than in lakes in the continuous permafrost zone (8 mg L$^{-1}$).

**4.5 Higher lake DOC concentrations in yedoma regions**

About 16 % of the 1,833 lakes of our dataset were located in regions with ice-rich syngenetic permafrost deposits (yedoma). The DOC concentration in lakes of these regions ranged from 1.7 mg L$^{-1}$ to 50.6 mg L$^{-1}$ with a median of 11.8 mg L$^{-1}$. The DOC concentrations in non-yedoma region lakes, comprising 79 % of the dataset, ranged from 0 mg L$^{-1}$ to 1,130 mg L$^{-1}$ and the median DOC concentration was 10.3 mg L$^{-1}$ which is significantly lower than in the yedoma region ($p < 0.05$). Our analysis shows a weak significant relationship of the lake surrounding deposit type and lake DOC concentration ($\rho = -0.2$; $p < 0.05$; Table 4; Fig. A2). Highest median DOC concentrations occur in lakes of areas with mountain alluvium and glacio-lacustrine deposits (15.2 mg L$^{-1}$, 15.5 mg L$^{-1}$). Lowest median DOC concentrations were found in lakes in areas underlain by bedrock, coastal and glacial deposits (2.6 mg L$^{-1}$, 4 mg L$^{-1}$ and 4 mg L$^{-1}$).

**4.6 Lower DOC concentrations in regions with low ground ice content**

Lakes of our dataset were located in regions of low, moderate, high and variable ground ice content (percentage of lakes: 36.5 %, 22.8 %, 25.4 % and 8.8 %, respectively). We found a weakly positive relationship between ground ice content and lake DOC concentrations ($\rho = 0.05$; $p < 0.05$; Table 4; Fig. A2). In regions of low ground ice content, the median amounts to 9.6 mg L$^{-1}$, compared to regions of moderate and high ground ice content with median DOC concentrations of 12.7 mg L$^{-1}$ and 11.4 mg L$^{-1}$, respectively.

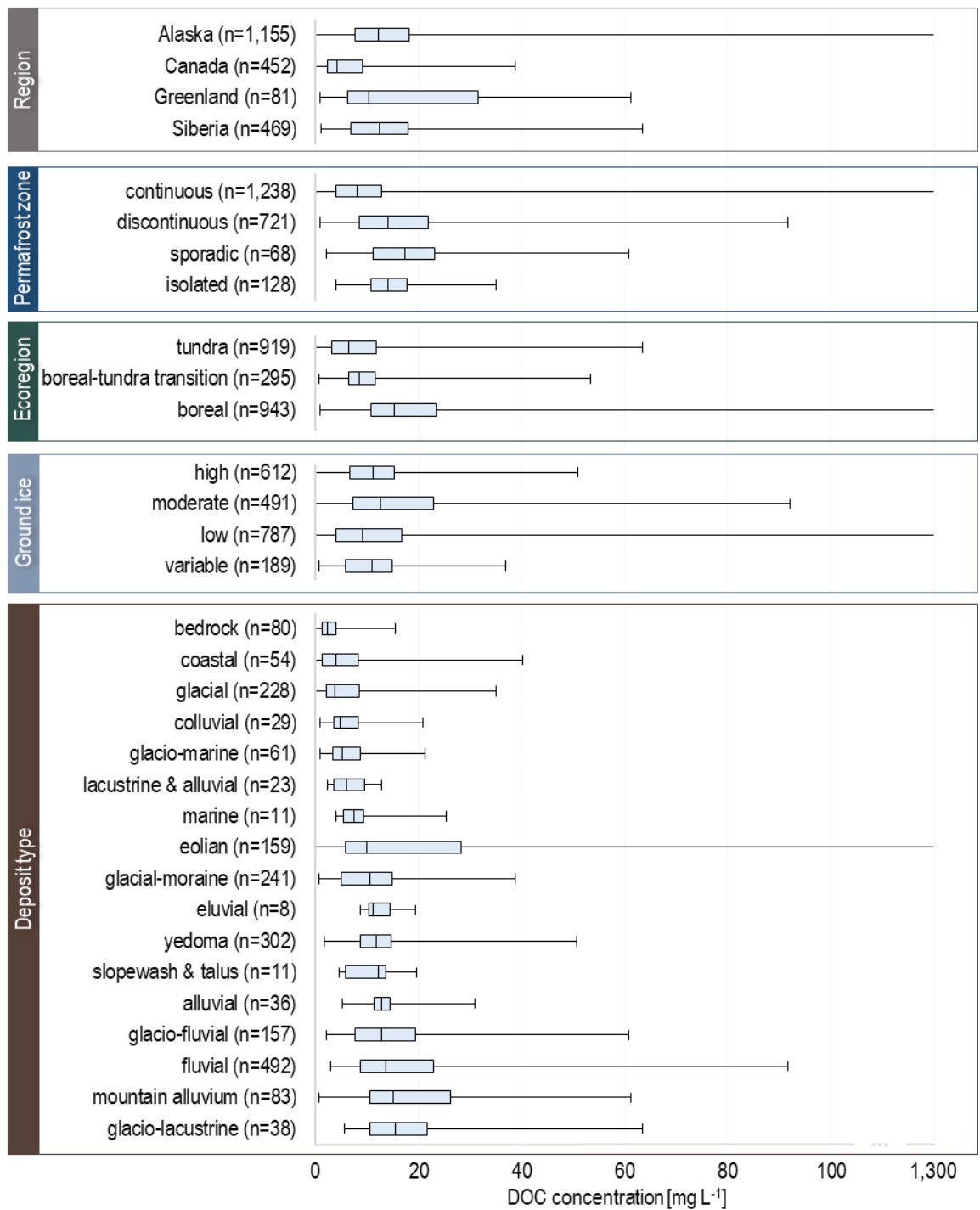

**Figure 3: Dissolved Organic Carbon (DOC) concentration (mg L$^{-1}$) of lakes in different Arctic regions, permafrost zones, ecoregions, ground ice content, and deposit types. Note that the x-axis is interrupted between 100 mg L$^{-1}$ and 1,300 mg L$^{-1}$ to visually capture the wide range of the DOC concentrations.**

## 4.7 Lake DOC and SOCC

We analysed the relationship between lake DOC concentrations and lake surrounding SOCC and found a weakly significant relationship for SOCC of the upper 100 cm ($\rho = 0.1$; $p < 0.05$; Table 4; Fig. A2). The significance of the relationship was getting weaker for SOCC in the upper 300 cm ($\rho = 0.09$; $p < 0.05$; Table 4, Fig. 4; Fig. A2).

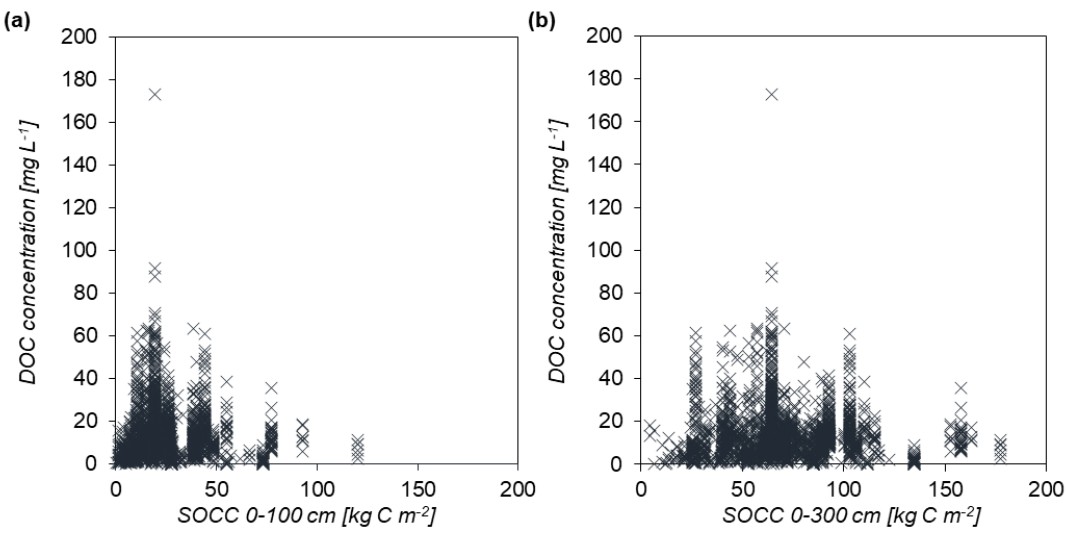

Figure 4: Scatterplots for lake DOC concentration and lake surrounding soil organic carbon content (SOCC) in a depth of 0 to 100 cm (a) and in a depth of 0 to 300 cm (b). To better visualize the relationship of both parameter we limited the y-axis to 200 mg L$^{-1}$. Three lakes with the DOC concentrations of 433 mg L$^{-1}$, 507 mg L$^{-1}$ and 1,130 mg L$^{-1}$ and SOCC of 19.7 kg C m$^{-2}$ in a depth of 0-100 cm and 64.6 kg C m$^{-2}$ in a depth of 0-300 cm are not included in this plot.

## 5 Discussion

### 5.1 Ecoregion zonation as key factor for pan-Arctic lake DOC

Our study shows the strongest significant relationships between lake DOC concentration and permafrost extent, ecoregion, and geographic latitude ($\rho = 0.31$; $\rho = 0.37$; $\rho = -0.3$). In contrast to Sobek et al. (2007), who assumed a strong relationship between lake DOC and soil OC, we found only a weak connection of lake DOC and surrounding SOCC. Our study provides an insight of potential sources of DOC in pan-Arctic lakes. We particularly found that lakes in the boreal forest region have higher DOC concentrations compared to tundra region lakes (Fig. 3). Soils of boreal forests are rich in organic material and microbial degradation is low (Sobek et al. 2007). In areas of boreal forest, the frost-free period is extended and the surface water can be in contact with soil C for a longer time resulting in higher DOC concentrations in boreal lakes. Previous studies confirm that vegetation is an important driver for DOC in permafrost catchments (Harms et al., 2016; Coch et al., 2019). Coch et al. (2019) found higher DOC concentrations in moss and plant rich Low Arctic catchments on Herschel Island in Northwest Canada compared to a High Arctic catchments at Cape Bounty in Northeast Canada. This relationship may explain high lake

DOC concentrations we found in the Yukon Flats in Interior Alaska, a study area in the boreal forest and dominated by white and black spruce (Halm & Griffith, 2014). In contrast, higher permafrost extent in high northern latitudes results in lower vegetation density and lakes are less connected and thereby hydrologically isolated, leading to overall lower DOC concentrations.

With climate change affecting northern ecosystem structures, a reduced permafrost extent (Vasiliev et al., 2020), shifting vegetation composition (Myers-Smith et al., 2011), and enhanced hydrological connectivity (Chen et al., 2014; Nitze et al., 2017) likely will impact lake DOC concentrations and associated biogeochemical fluxes (Sobek et al., 2005). For example, enhanced DOC concentrations in a lake provides an increased basis for the mineralization of DOC through photo oxidation and by microbial activity, which may result in higher $CO_2$ emissions from these lakes. In our first pan-Arctic assessment of

DOC in lakes of the permafrost region we found that DOC concentrations in lakes become significantly higher along an ecoregion gradient transitioning from tundra zone to the tundra-boreal transition zone to the boreal forest zone. In addition, DOC concentrations are overall higher in permafrost zones that are less continuous. Both trends suggest that climate change, projected to result in an expansion of the boreal forest northwards into the tundra zone and a decrease in permafrost continuity will likely result in higher DOC concentrations in lakes of these regions. Moreover, permafrost loss and a shift of the boreal

forest ecoregion might lead to more connected lakes and thus an increase of allochthonous DOC in lakes. This in turn, may result in higher $CO_2$ emissions from lakes to the atmosphere.

**5.2 Pan-Arctic lakes in a global view of lake DOC**

The median DOC concentration of our dataset (10.8 mg L$^{-1}$) is almost three times higher than the value (3.88 mg L$^{-1}$) found by Toming et al. (2020), who studied global lakes with a surface area larger than 0.1 km$^2$. Our study across the Arctic shows

a high variation of lake DOC concentration. Canada and Greenland had the lowest median DOC concentration with low inter-site variation (Fig. 3) compared to the high variability observed in Alaska and Siberia. Whereas the Canadian and Greenlandic regions were affected by past glaciation, the majority of the Alaskan and Siberian sites were not glaciated and are characterized by extensive low-lying wetlands. Though, we found a weak significant relationship between lake DOC concentration and lake surrounding deposit type, we found the lowest DOC concentrations in lakes surrounded by glacial and bedrock deposits

(Fig. 3). In our dataset, these deposit types are mainly located in the former glaciated Canadian Arctic. Sepulveda-Jauregui et al. (2015) found a higher DOC content in yedoma lakes, analysing $CO_2$ emissions from 40 lakes of a north-south transect in Alaska covering all permafrost types. So, we compared the DOC concentration in lakes of the yedoma region and the DOC concentration in lakes in non-yedoma regions, comprising 79 % of our dataset with different deposit types and including lakes with the four highest DOC concentrations in Interior Alaska characterized by fluvial deposits, eolian deposits and mountain

alluvium deposits. We found significantly higher DOC concentrations in yedoma lakes compared to non-yedoma lakes. This might be attributed to the mobilization of old labile yedoma carbon by thermo-erosion along rapidly expanding lake shores and thermokarst processes (Strauss et al., 2017). We assume that yedoma lake generation is influencing yedoma lake DOC. The formation of yedoma lakes, due to deep thermokarst subsidence, results in deep and often closed basins (Morgenstern et

al., 2011). As result of the missing lake connectivity, DOC is locked in the lake, originating partially from eroding organic-rich yedoma deposits (Strauss et al., 2017), melting yedoma ice wedges (Fritz et al., 2015) and from the active layer. Further the lower lake connectivity might prevent flushing of yedoma thermokarst lake water with river water and snowmelt water. Hence, we assume that yedoma thermokarst lakes are more likely to have elevated DOC concentrations than other more

connected lakes as well as well-mixed larger and shallower lakes, where photodegradation plays an important role, associated with lower lake DOC concentration. However, to determine the DOC source in yedoma lakes radiocarbon dating of each sample would be necessary.

While we showed that lake DOC concentration is influenced by permafrost extent and type of ecoregion they do not explain all of the variability in the dataset. Additional factors are regulating DOC. For example, air temperature, precipitation and solar

radiance have an influence on surface water DOC concentration (Cole et al., 2002; Molot et al., 2005; Anderson & Stedmon, 2007). Anderson & Stedmon (2007) analysed lakes in low Arctic Greenland and found highest lake DOC concentrations in areas of low precipitation and low discharge. In those areas, evaporation is high leading to higher DOC concentrations. For our database, the role of evaporation may also explain the high DOC concentrations of lakes in the Yukon Flats in Interior Alaska. Here, the lakes are less hydrologically connected and the region is very arid, allowing evaporation-driven concentration

of DOC (Johnston et al., 2020).While we found that lake latitude is correlated to lake DOC concentration, we did not investigate lake altitude. Sobek et al. (2007) and Toming et al. (2020) found for their global lake databases that lake altitude is another important indicator for lake DOC, with lake DOC concentrations being lowest in areas of high elevation.

## 5.3 The complexity of lake DOC regulation

Analysis of our dataset with available pan-Arctic data have shown significant relationships between ecoregion and lake DOC

concentration, between geographical latitude and DOC concentration, and between permafrost extent and DOC concentration, even if these relationships are generally weak. Other studies suggest additional parameters influencing lake DOC concentration. For example, Xenopoulos et al. (2003) analysed catchment characteristics of lakes and found that lake perimeter and the proportion of the watershed occupied by wetlands are strongest predictors for DOC in lakes of temperate forests. On a global scale, lake elevation and the proportion of wetlands in a watershed are strongest predictors for lake DOC. Tranvik et

al. (2009) described that lake area, which is connected to lake volume and water retention time, might be negatively correlated in regional studies but that it is not an important DOC predictor in a global view. The fact that the majority of predictors for lake DOC differ in regions demonstrate the complexity of the regulation of DOC concentration in lakes. However, several of these parameters are not included in our study, which could be a cause for the often only weak relationships found in our analysis. As a result of limited data availability on detailed hydrological catchments of northern lakes, the hydrological

connectivity (vertical and lateral) is also not included in our analysis. However, it is known for example that less allochthonous DOC is transported to a hydrologically isolated lake than to a connected lake (Bogard et al., 2019). In arid regions with rather isolated lakes, such as in the Yukon Flats in Interior Alaska, evapoconcentration of DOC plays an important role (Johnston et al., 2020). Water bodies with highest DOC concentrations in the Yukon Flats have a water depth less than 1 m. Studies in

West-Siberia showed, that ponds receive the highest impact of allochthonous input due to the high ratio of lake drainage area vs. small water volumes. This results in short water residence time leading to highest concentrations of DOC (Shirokova et al., 2013; Manasypov et al., 2014, 2015). In addition to allochthonous DOC, autochthonous DOC, including phytoplankton productivity as well as heterotrophic bacterioplankton respiration processes (Chupakov et al., 2017), is influencing the DOC concentration, especially in lakes with low connectivity. For lakes in the Yukon River Basin, Bogard et al. (2019) described a minor importance of allochthonous DOC in lakes and highlighted the carbon fixation from atmospheric $CO_2$.

Beside our analysis of temporal variability of a subset of our dataset, the sampling month of each sample was not included in the statistical analysis of our pan-Arctic dataset, which may result in uncertainties due to variations in lake DOC concentration over the ice-free period. For Qeqqata, Greenland, higher DOC concentrations were found in samples collected in April (under ice) and August compared to June samples. In winter, nutrients as well as DOC do concentrate in lakes (Manasypov et al., 2015; Vonk et al., 2015; Grosbois et al., 2017), resulting in higher DOC concentrations in under-ice samples from April. The spring flood transports large amounts of allochthonous DOC to the lakes, concentrating them with DOC resulting in higher lake DOC concentrations in spring (Manasypov et al., 2015). During summer, in this region, characterised by low precipitation, evapoconcentration is a major cause for increasing DOC concentration (Anderson & Stedmon, 2007). Considering a seasonality of DOC concentration in our dataset, we found two different patterns in two different study sites. This highlights the complexity of regulators and mechanisms of the DOC concentration in a lake over a season.

The influence of biological, hydrological, climatic and topographical parameters on the DOC concentration of a lake clearly is very complex. Whereas our pan-Arctic dataset provides first insights of the relationship between some environmental parameter and lake DOC concentration, regional studies are necessary to understand these complex mechanisms and to determine DOC predictors, which may differ regionally.

## 5.4 Challenges of a pan-Arctic DOC assessment

Our synthesis shows a wide range of DOC concentrations in Arctic permafrost region lakes. An important uncertainty factor for analysing lake DOC concentration in a pan-Arctic context is the still limited amount of lake DOC data compared to the exceptional large number of lakes. This region hosts the most lake-rich landscapes on earth (Lehner & Döll, 2004) and their geologic and hydrologic origins are diverse (Pienitz et al., 2008; Vincent & Laybourne-Parry, 2008; Grosse et al., 2013) but often connected to paleogeographic and cryosphere processes that are differing substantially from the world's other lake regions (Smith et al., 2007; Brosius et al., 2021). Lakes in our synthesis dataset were sampled over the past 40 years (Fig. A1). Since then, environmental conditions in some study areas may have changed due to the accelerating climate change. For example, thermokarst lakes are very dynamic and some lakes that were sampled 30-40 years ago may be now be completely drained and thus no longer exist. Other environmental characteristics in catchments such as permafrost extent, vegetation cover, or runoff dynamics may have changed over time thereby also affecting lake DOC concentration. The remoteness of many lakes in the Arctic results in multiple challenges to spatially and temporally representative sampling. For example, multi-temporal sampling of Arctic lakes is still very rare and limits our insights in the seasonal and long-term dynamic of lake DOC

of many Arctic lake types. To our best knowledge, there are no long-term lake DOC studies available for the Arctic that would help understanding decadal-scale DOC changes and trends and possible correlations with ongoing Arctic change. However, seasonal fluctuations were studied for a small subset of lakes in our dataset (Qeqqata, Greenland).

Further uncertainties result from still rather coarse-resolution environmental data layers for the pan-Arctic such as permafrost,
ground ice content, soil organic carbon, ecoregion, as well as the sparseness of high-resolution climate data. New remote sensing and numerical modelling-driven approaches to create spatially homogeneous datasets for this large region may provide a much better base for future analyses of lake DOC and its correlation with environmental factors. For example, pan-Arctic remote sensing of permafrost region disturbances (Nitze et al., 2018) may allow correlation of lake DOC data with the processes of rapid permafrost degradation, or global studies of remotely sensed lake abundance and change (Pekel et al., 2016)
may help to understand the dynamical aspects of lake DOC. To quantify the permafrost region lake DOC pool, an assessment of the volume of the diverse lake types in the Arctic is needed.

## 6 Conclusion

DOC is one of the main C fractions in lakes contributing to the greenhouse effect as part of the global C cycle. This first pan-Arctic assessment provides linkages between DOC concentrations and the environment of 1,833 lakes in permafrost regions
of Alaska, Canada, Greenland and Siberia. Our study compares DOC concentrations of lakes in the permafrost region with different permafrost extent, tundra and boreal forest ecoregions, regions of different deposit types, areas with high, moderate, low and variable ground ice content and different SOCC in the upper 3 m. In these areas, we found a wide range of DOC concentrations from 0 to 1,300 mg $L^{-1}$ with the highest concentrations in lakes in the Yukon Flats in Interior Alaska and lowest in the North Slope in Arctic Alaska and the Canadian Arctic Archipelago. We identified a significant relationship of lake DOC
and the ecoregion and we found increasing lake DOC with increasing vegetation from tundra to boreal forest and decreasing latitude and permafrost extent. We conclude for our dataset that ecoregion zonation is the most important driver for lake DOC concentration in the pan-Arctic region. Nevertheless, the regulation of lake DOC concentration is complex and some DOC predictors, such as hydrological connectivity, water retention time and topography, were not included in our analysis due to the lack of appropriately detailed pan-Arctic datasets for these parameters. However, our study of pan-Arctic lake DOC
concentration in permafrost regions provides a first broad overview of the connections between lake DOC and lake environment and forms a basis for further detailed analysis. So, the new PerL-DOC database will be useful for quantification of C pools and fluxes from freshwater bodies across the Arctic.

## Appendix A: DOC analysis at Alfred-Wegener-Institute (AWI)

For DOC analysis of 246 samples collected by authors from AWI, 20 ml of the sample was filtered through a 0.7 µm pore size glass fiber filter, preserved with 20-50 µl of 30 % hydrochloric acid (HCl) and sent to AWI in Potsdam, Germany, for

laboratory processing. We then treated the samples with high-temperature catalytic combustion. For the quality control during the measurement and validation of the results, standard samples with known concentrations of DOC and blank samples of ultrapure water were added to the sample set. The direct method or so-called NPOC-method (Non-Purgeable-Organic-Carbon) was used to determine the DOC concentration. We filled 9 mL of the sample into a 9 mL glass vial, sealed each vial with an

5   aluminium foil, and placed them in the vial rack of 'Shimadzu TOC-VCPH'. During measurement, the samples were acidified with hydrochloric acid to a pH value of 2-3 and afterwards treated with oxygen gas, which eliminated inorganic C by conversion to $CO_2$. In the next step, NPOC passes the catalyst, where it heats up to 680 °C and the $CO_2$ passes the NDIR detector (Non Dispersed InfraRed). The NDIR detector measures the concentration and related software calculates the average of up to five measurement procedures of each sample (Manual Shimadzu/TOC-V, 2008). The DOC concentration was recorded

10   in mg $L^{-1}$.

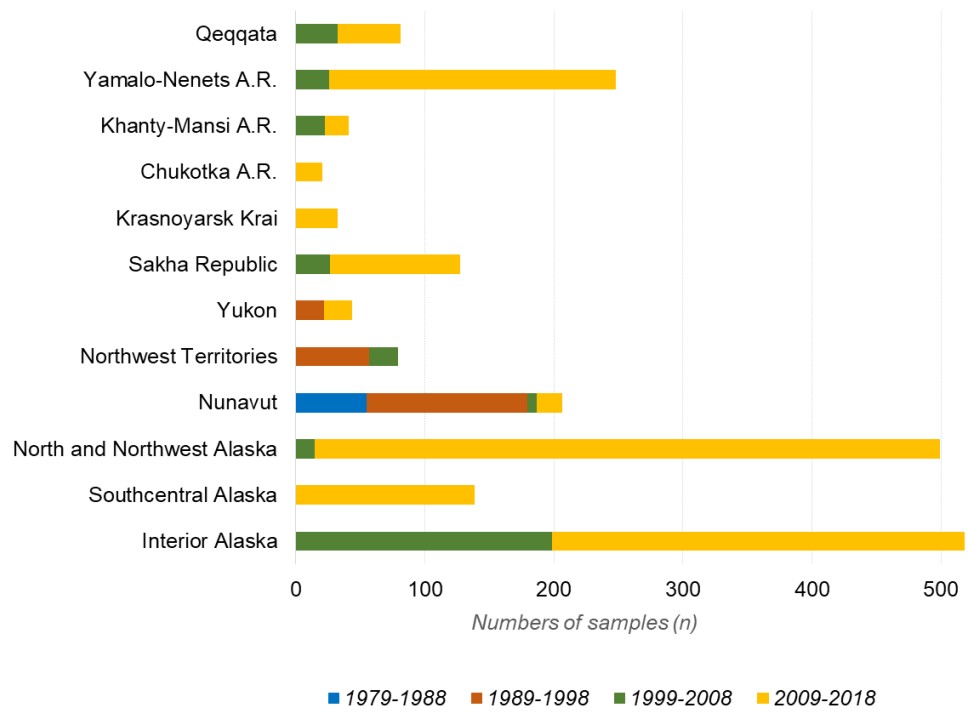

**Figure A1: Histogram of numbers of samples in each study area per decade of sample collection. Since an exact allocation of the sampling year was not possible for 113 samples, these are missing in this figure for study areas Nunavut and Manitoba, Canada.**

**Table A1: Parameters for lake analyses.**

| General | Lake Name/ID | |
|---|---|---|
| | Sample date | |
| | Location | Latitude and Longitude. |
| | Study Area | Qeqqata, Yamalo-Nenets Autonomous Region, Khanty-Mansi Autonomous Region, Chukotka Autonomous Region, Krasnoyarsk Krai, Sakha Republic (Yakutia), Yukon, Northwest Territories, Nunavut, Manitoba, North and Northwest Alaska, Southcentral Alaska, Interior Alaska. |
| | Study Site | Qeqqata, Yamalo-Nenets Autonomous Region, Khanty-Mansi Autonomous Region, Chukotka Autonomous Region, Khatanga, Lena Delta, Kolyma, Herschel Island, Yukon Coastal Plain, Whitehorse transect, Mackenzie Delta, Tuktoyaktuk transect, Yellowknife-Contwoyto transect (Northwest Territories), Yellowknife-Contwoyto transect (Nunavut), Canadian Arctic Archipelago, Repulse Bay, Arviat area, Baker Lake area, Rankin Inlet area, Churchill area, North Slope, Kobuk Delta, Kobuk Valley national Park, Noatak Delta, Noatak National Preserve, Baldwin Peninsula, Selawik Delta, Bering Land Bridge National Preserve, Seward Peninsula, Wrangell-St. Elias National Park and Preserve, Denali National Park and Preserve, Yukon Flats, Yukon-Charley Rivers National Preserve. |
| | Permafrost zone | Continuous, Discontinuous, Isolated, Sporadic. |
| | Ecoregion | Tundra, tundra-boreal transition, boreal forest |
| | Data Source | Reference or sample collector. |
| Hydrochemistry | DOC | In mg L$^{-1}$. |
| Geology | Deposit type | Coastal, eolian, yedoma, glacial-moraine, glacio-lacustrine, glacio-fluvial, fluvial, mountain alluvium, glacio-marine, glacial, bedrock, colluvial, alluvial, lacustrine and alluvial, marine, eluvial, slopewash. |
| | Ground Ice Content | Low, moderate, high, variable. |
| | Soil Organic Carbon Content | Kg C m$^{-2}$, in 0 – 100 cm, 100 – 200 cm, 200 – 300 cm, and summed 0 – 300 cm of upper soil. |

**Table A2: Overview of median DOC concentration according to sampling months, where sampling month could be clearly identified.**

| | Study area | Median DOC concentration [mg L$^{-1}$] | | | | | | |
|---|---|---|---|---|---|---|---|---|
| | | April | May | June | July | August | September | October |
| Greenland | Qeqqata | 33.1 | | | 10.2 | 25 | | |
| Siberia | Yamalo-Nenets Autonomous Region | | | 18.1 | 29.6 | 13.9 | | |
| | Khanty-Mansi Autonomous Region | | | | 14.1 | 10.9 | | |
| | Chukotka Autonomous Region | | | | 9.6 | | | |
| | Krasnoyarsk Krai | | | | 8.3 | 9.8 | | |
| | Sakha Republic (Yakutia) | | | | 8.1 | 10.2 | 7.8 | 15.3 |
| Canada | Yukon | | | | 12.9 | | | |
| | Northwest Territories | | | | 8.5 | | 13 | |
| | Nunavut | | | | 2.2 | 3.8 | 2.6 | |
| Alaska | North and Northwest | | | 4.5 | 9.1 | 8.2 | 8.1 | |
| | Southcentral | | 15 | 13.6 | 12.5 | 16.3 | 15 | |
| | Interior | | 12.7 | 18.4 | 12.6 | 20.3 | 21.3 | |

**Table A3: DOC concentrations in study sites.**

| | Study area | Study site | n samples / n lakes | DOC concentration [mg L$^{-1}$] | | |
|---|---|---|---|---|---|---|
| | | | | range | mean | median |
| **Alaska** | | | **1,165/903** | **0 – 1,130** | **17** | **12.3** |
| | *North and* | | *499/397* | *0 – 53.3* | *9.6* | *8.6* |
| | *Northwest* | North Slope | 64/16 | 0 – 20.8 | 3 | 1.7 |
| | | Kobuk Delta | 14/14 | 4.3 – 40.1 | 11.2 | 8.1 |
| | | Kobuk Valley National Park | 157/112 | 2.9 – 53.3 | 9.7 | 7.9 |
| | | Noatak Delta | 3/3 | 7.9 – 12.2 | 9.8 | 9.3 |
| | | Noatak National Preserve | 114/107 | 0.7 – 20.9 | 9.6 | 9.2 |

| | | | | | | |
|---|---|---|---|---|---|---|
| | | Baldwin Peninsula | 3/3 | 12.6 – 35.4 | 20.3 | 12.8 |
| | | Selawik Delta | 4/4 | 8.4 – 11.4 | 10.4 | 11 |
| | | Bering Land Bridge National Preserve | 121/119 | 4.7 – 25.8 | 11.2 | 10.7 |
| | | Seward Peninsula | 19/19 | 1.4 – 38.3 | 16.1 | 16.4 |
| | *Southcentral* | *Wrangell-St. Elias National Park and Preserve* | *138/126* | *0.8 – 36.8* | *14.1* | *13.8* |
| | *Interior* | | *528/380* | *1.4 – 1,130* | *25* | *16.8* |
| | | Denali National Park and Preserve | 257/161 | 1.4 - 35 | 12.6 | 12.4 |
| | | Yukon Flats | 150/140 | 10.2 – 1,130 | 48.7 | 30.3 |
| | | Yukon-Charley Rivers National Preserve | 121/79 | 5 – 66.7 | 23.8 | 22.7 |
| **Canada** | | | **452/444** | **0 – 38.7** | **6.6** | **4.2** |
| | *Yukon* | | *54/54* | *3.1 – 38.7* | *15.4* | *14.7* |
| | | Herschel Island | 20/20 | 5.4 – 38.7 | 17.6 | 17.3 |
| | | Yukon Coastal Plain | 12/12 | 5.6 – 25.4 | 14.6 | 12.7 |
| | | Whitehorse transect | 22/22 | 3.1 – 35.1 | 13.9 | 12.9 |
| | *Northwest* | | *79/79* | *1.7 - 30* | *10.2* | *9.1* |
| | *Territories* | Mackenzie Delta | 22/22 | 6.8 - 30 | 13.4 | 13 |
| | | Tuktoyaktuk transect | 37/37 | 3.9 – 29.9 | 11.3 | 10.1 |
| | | Yellowknife-Contwoyto transect | 20/20 | 1.7 – 9.1 | 4.7 | 4.3 |
| | *Nunavut* | | *302/294* | *0 – 31.9* | *3.9* | *2.9* |
| | | Yellowknife-Contwoyto transect | 4/4 | 1.6 – 2.7 | 2.2 | 2.3 |
| | | Canadian Arctic Archipelago | 220/212 | 0 – 31.9 | 3.6 | 2.5 |
| | | Repulse Bay | 6/6 | 2 – 5.4 | 3.8 | 4.2 |
| | | Arviat area | 25/25 | 2.6 – 11.7 | 6.5 | 5.8 |
| | | Baker Lake area | 28/28 | 2.2 – 5.4 | 3.2 | 3.1 |
| | | Rankin Inlet area | 19/19 | 2.7 – 17.4 | 5.9 | 4.5 |
| | *Manitoba* | *Churchill area* | *17/17* | *2.7 – 21.2* | *9.1* | *7* |
| **Greenland** | **Qeqqata** | | **81/59** | **1 – 61.3** | **18.5** | **10.3** |
| **Siberia** | | | **469/427** | **1.1 – 63.4** | **14.4** | **12.4** |

| | | | | | |
|---|---|---|---|---|---|
| *Yamalo-Nenets Autonomous Region* | | | *249/249* | *3.2 – 63.4* | *18.1* | *15.6* |
| *Khanty-Mansi Autonomous Region* | | | *41/41* | *5.8 – 36.1* | *13.7* | *11* |
| *Chukotka Autonomous Region* | | | *20/20* | *1.1 – 19.6* | *9.5* | *9.6* |
| *Krasnoyarsk Krai* | *Khatanga* | | *32/32* | *2.3 – 19.4* | *8.3* | *8.3* |
| *Sakha Republic* | | | *127/85* | *2.4 – 33.3* | *9.6* | *9.8* |
| *(Yakutia)* | Lena Delta | | 100/66 | 2.4 – 33.3 | 8.6 | 8.2 |
| | Kolyma | | 27/19 | 5.3 – 22.7 | 13.2 | 12.6 |
| **Total** | | | **2,167/1,833** | **0 – 1,130** | **14.1** | **10.8** |

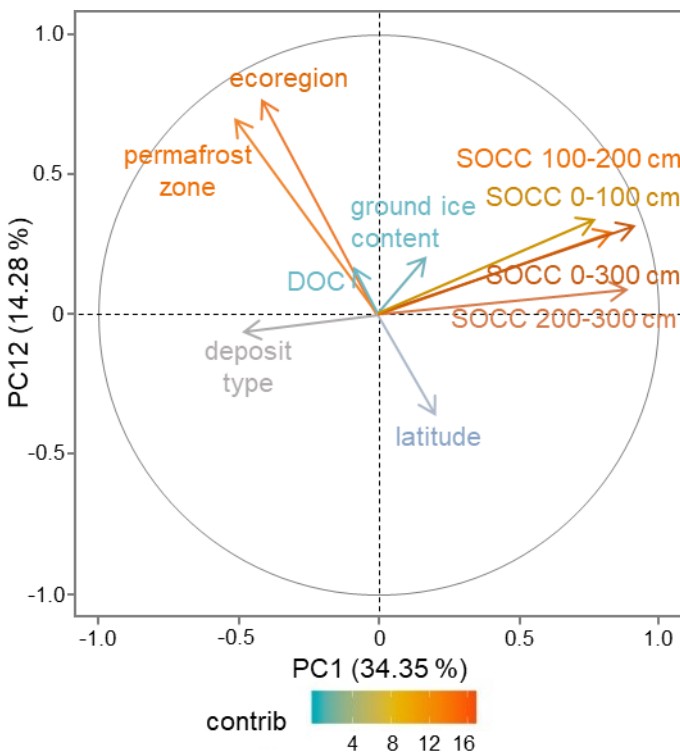

**Figure A2: Principal component analysis showing a variables factor map with a color gradient showing the contribution to the plane construction. The first two principal components explained 34.35 % (PC1) and 14.28 % (PC2) of the variance in the analysed parameters (DOC, latitude, permafrost zone, ecoregion, ground ice content, deposit type, SOCC). The scores of PC1 had positive loadings with SOCC in all depths and negative loadings with deposit type, while PC2 scores had negative scores of latitude and positive scores of DOC, permafrost zone, ecoregion and ground ice content.**

## Data availability

The Permafrost Region Lake-DOC version 1 (PeRL-DOCv1) dataset was submitted to PANGAEA, is actually in the review process and will be available soon.

## Author contribution

LS lead the DOC data collection and synthesis, created the database and led the writing of the manuscript. LS and GG conducted the literature search for appropriate DOC datasets. LS, CC, AM, JB, MF, UH, KSL, BH, JL, KWA, BJ, KF and GG contributed so far unpublished DOC data for this study. LS and CC performed statistical analyses. All co-authors contributed to the writing.

**Competing interests**

The authors declare that they have no conflict of interest.

**Acknowledgements**

This study was supported by a PhD stipend of the University of Potsdam awarded to LS and the ERC PETA-CARB project
(338335). US National Science Foundation awards OPP-1107481 and OPP-1806213 contributed to this research. The author
team would like to thank all colleagues being involved in sample collection in the field. Fieldwork in the Alaskan National
Parks was funded by the National Park Service, Central Alaska Network Inventory and Monitoring Program. Thanks to
Christopher Arp for contributing lake DOC data for the Alaska North Slope. Further, we thank Sebastian Laboor, Paul
Overduin and the laboratory staff of AWI Potsdam, in particular Antje Eulenburg. Samples in Central Yamal were collected
within the field campaign of the Earth Cryosphere Institute (TyumSC SB RAS) in 2015 and processed at Otto-Schmidt
Laboratory in Saint Petersburg.

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
