# Peer review of "First Pan-Arctic Assessment of Dissolved Organic Carbon in Lakes of the Permafrost Region"

_Biogeosciences, 2020_

## Referee Comment (RC1) · Anonymous Referee #1 · 9 Dec 2020

General Comments: The authors have presented a synthesis of a very large data set of DOC concentrations from permafrost affected lakes spanning the entire Arctic and a straight forward, but yet effective regional scale analysis of the landscape controls on the DOC concentrations in these lake.

The authors have identified the data gap and clearly spelled out the objectives of the study They have done a commendable job at gathering this very impressive and novel data set, they have provided a detailed summary of the temporal and spatial nature of the data, and have applied valid statistical tools to investigate and identify the role that a range of landscape variables play in controlling the DOC concentrations in Arctic/permafrost lakes. The authors have also done a very good job at addressing the uncertainties and challenges with the data in the discussion.

On it's own the act of gathering and describing this data set constitutes a valuable contribution, and the analyses conducted and conclusions drawn add significantly the value of that contribution. The research presented is well within the scope of the journal and should be of interest to a wide audience. As such I believe the work is definetly worthy of publication. I do feel that the data have been under-utilized in a few respects, and/or there are some analyses that could be added to address some of the issues of uncertainty and the variability in the data and strengthen the paper.

I detail the suggestions in the specific comments below. I view these as minor to moderate revisions, as they require some additional analyses or description of the data, but I do not believe they will change the outcomes or conclusions that are reached.

Specific Comments:

Title – I suggest a couple minor changes. First I would remove "-Region" from the title - simply state Permafrost Lakes.

Second I would advise the authors to consider editing the title slightly to specify the nature of the "Assessment" conducted.

Perhaps: "First Pan-Arctic Assessment of landscape characteristics as controls on Dissolved organic carbon in Permafrost Lakes"

Or "First Pan-Arctic Assessment of environmental parameters as drivers for Dissolved organic carbon concentrations in Permafrost Lakes" Or along these lines.

Study Area: The authors spend several paragraphs on pages 3-5 describing the percentages of the lakes that are from the different Arctic regions and the different ecozones etc. Although the authors do provide the lake/sample numbers as n values in Table 2 and Figure 3, because there is a strong spatial bias (with more than half the lakes in Alaska) I suggest that a histogram be included, to better illustrate the geo-

graphical distribution of the lakes. This could be included as an insert for example in Figure 1 (using this regional description).

The other aspect of the data that is not clearly illustrated is the timeframe of collection. It would help the reader understand the data, if there was some illustration of the number of samples/lakes that were taken from the various years (e.g. how many samples are from 1979-1985 in Nunavut?). I do not think it would require much effort to generate a temporal histogram (with the number of samples from the various years and quarters).

The authors need to address the potential impact of including samples from nearly 40yr span of time could have on the analyses, especially given that climate and permafrost has been changing dramatically (and at different rates) across these ecoregions during this time. For example this long time frame of the sample period could play a role in the relationship between DOC and latitude. Is it possible the relationships might be more robust if the authors limited themselves to data from the last decade or two? The temporal histogram suggested above would provide a means to speak to this issue.

Results: Temporal Variability of the data set.

I like that the authors indicate the nature of the seasonality of the sampling. Although section 4.1 breaksdown how many lakes were sampled multiple times, and how the DOC concentrations varied in terms of the overal range and trends in concentrations of DOC (i.e. across all lakes) through different seasons, I feel what is lacking is an assessment of the degree to which DOC concentrations vary in individual lakes over the year or seasons. i.e. is the range of variability within a year/season in a given lake greater than the variability between lakes for this subset? I suspect this seasonal variability is minimal, however if seasonality is important then the authors would need to consider limiting the data set to lakes sample from a consistent time of the year?

Although the temporal subset is small relative to the whole data set, there are 81 lakes and 266 samples, which is by most any measure still a substantial data set. There is

**BGD**

likely sufficient data within this to provide some assessment of the relative impact of both the seasonality on the data.

I didn't see a data availability statement. Even if this is not a requirement of the journal, it is critical and needs to be included – perhaps in the results or cited as a data archive?. Readers are going to want to know how or where they might be able to access this valuable data set.

Technical corrections:

P1 L29 Edit: Our synthesis shows a significant relationship of lake DOC concentration and ecoregion of the lake. *cut "of" insert "between" , insert "lake" ahead of "ecoregion" and cut "of the lake"

P1L32 Compared to previous studies we found a weak significant relationship of soil organic carbon content. . .. *cut "of" insert "between"

P7 Line 21 - add "data" to the end of the section title

P13 L16 change "with a surface are " to ". . .surface area"

P.21 L20 – check reference volume number "Limnol. Oceanogr., 9999"?

---

## Referee Comment (RC2) · Anonymous Referee #2 · 7 Jan 2021

The manuscript presents a newly assembled, Global dataset of DOC concentrations from northern lakes. It is a very nice dataset that is clearly valuable. The mapping of patterns of DOC throughout figures 1 to 3 and tables 1 to 2 are also very useful context for readers. On this basis, I feel the manuscript has great potential to make an important contribution to the field. However, while the dataset is powerful, and geographic patterns are interesting, the paper is lacking in mechanistic insight, with a number of important concepts overlooked. These issues should be resolved before the manuscript could be considered fit for publication. I outline my concerns below, and hope that the authors find this evaluation constructive and useful.

[Figure]

General comments:

1. I think that the core message of the paper needs to change from one of strongly predicting lake DOC, to stating that predicting DOC patterns at the Global scale is complex, and has resulted in weak relationships with the individual predictors at hand. Throughout, the authors claim that lake DOC strongly depends on environmental properties. I agree, yet I don't think that the results presented here have led to this conclusion. Disregarding p values (which are uninformative due to their dependence on sample size), the correlation strength for every environmental parameter versus DOC is weak, and weaker than those for basic categorical groupings (region/zone) and latitude, which themselves are marginal in strength. The fact that these categorical and spatial variables remain stronger predictors than environmental measures (ice content, ground type, soil C content) tells us that the major drivers of lake DOC are not captured in this dataset (see next point on which predictors I mean). That isn't surprising, and speaks to the complex regulation of lake DOC concentrations. To fix this problem, I think that the messaging of the paper needs to change to emphasize the weakness of the strength of these individual predictors, and the complex control of lake DOC content has to be discussed throughout and emphasized. Throughout, I feel that readers are presented with an over-simplified view of the regulation of DOC concentrations.

2. The paper overlooks the most important mechanisms controlling DOC cycling, i.e., the roles of hydrology and geomorphology in structuring the delivery of DOC to lakes, and water residence time and allochthonous DOC processing. This may be a main reason for such weak relationships presented in their correlation table. Many other studies show that these factors critically shape lake DOC cycling, and without this information brought in to the analyses, the authors are likely missing a big part of the mechanistic story. For instance, the first paragraph of the discussion talks about vegetation density, soil C contact with water, and the effects of permafrost extent on vegetation. It really overlooks these major factors that are very much a part of that story, thereby making the mechanistic explanations in the paper incomplete. To fix this, the authors could improve the analyses by bringing in new datasets on hydrology, topography (catchment slope), and estimates of water residence time (even for a subset of the lakes) to explore this. At the very least, the importance of these factors needs to be better incorporated into the narrative of the paper, from start to finish.

3. The paper overlooks the role of autochthonous DOC production, and its importance in structuring patterns of DOC within and among regions. It is clear from past work using a bunch of different approaches in different regions, that the sources and composition of DOC ranges widely, and autochthonous sources can be very important in many lakes (e.g., Tank et al. 2011 L&O; Osburn et al. 2017 JGR-B; Osburn et al. 2019 L&O Lett.; Johnston et al. 2020 L&O). A lot of the high-DOC lakes in some regions are indeed rich in autochthonous DOC, so this source input may drive some of the most extreme observations in the current dataset, especially in the Yukon Flats region the authors highlight, which is discussed in that Johnston et al. 2020 paper. The importance is probably lower in some regions than others, and depends on hydrologic connections to the terrestrial landscape and other factors (getting back to comment 2 above). Overall, this is an important factor structuring lake DOC patterns that should at the very least be incorporated throughout the paper, where mechanistic inferences are made. To go a step further, the authors could look at a subset of the lakes for which published organic matter properties are available (optical, isotopic, elemental ratios, or other, depending on what is out there).

Specific comments: P1/L32 – How does this relationship compare, specifically?

P1/L34 – I don't think you demonstrate a strong dependence here, see general comment 1.

P2/L4 – abbreviate carbon as C after first mention. Comment applies throughout.

P2/L11 – regional warming: is this air or soil temperature? Be specific.

P2/L28 – Which lake-based process, specifically?

P2/L30 – Last sentence needs improving: Which flux? Lakes, streams, what? The paragraph could also be improved by adding a conclusion sentence identifying what the major knowledge gap is here.

P3/L1-9 – This paragraph is somewhat off topic. What does it have to do with lake DOC? Either revise it to link to the topic, or cut it out.

P3/L18 – Lake number or lake area?

P3/L25 – Throughout: Cold/Very Cold/Cool – give us some meteorological values to help understand what these designations mean.

P7/L21 – This section provides little insight. As it is a paper on global patterns, the seasonality seems off topic. What's more, there is no context to evaluate DOC patterns in a meaningful way (which lakes, which regions, etc.). Consider removing this.

P8/L7 – Where is table A2? Why not cite Fig. 2 & 3 with statistics reported to confirm this statement. Comment applies throughout paper about table A2.

P10/L1 – Instead of fig. 2a, why not cite Fig 3 here? Makes the point better.

P10/L5-6 – Why a new paragraph? Same topic/theme.

P11 – Fig. 3 – Where are the statistics? ANOVA and post hoc tests for each panel?

P12/L14-16 – Adding scatterplots and show us the data – the table is good but not enough.

P12/L24 – Does the study provide this insight? I don't think so, since the relationships are quite weak. See general comments for suggestions to improve this section.

P13/L4 – Should the word 'catchment' be plural?

P13/L7 – What about the effect on hydrologic connectivity caused by permafrost extent? Not just terrestrial veg distribution, but also consider the role of permafrost in disconnecting lake surface waters from hydrologic flowpaths that deliver DOC.

P13/L9 – Odd conclusion sentence. Seems unrelated. Consider revising.

P13/L12 – Weak cross-regional correlations don't really tell us about climate change impacts. Consider revising.

P13/L13 – The discussion throughout this last paragraph assumes all DOC comes from terrestrial environments. Not accurate. See general comment #3.

P13/L20 – "Our results" – be specific here and say which results.

P13/L20 to 21 – You do not have the data to infer this mechanism. Revise language here. Also, I do not think that this conclusion is consistent with the comparison of DOC concentrations by deposit type categories in Fig. 3. Yedoma type is not the highest DOC concentration there.

P13/L23-24 – You can't assume this without hydrologic or other information.

P13/L25 – This could be revised to be more mechanistically insightful. Wouldn't we expect lower delivery of allochthonous DOC to these lakes where permafrost limits hydrologic exchange into lakes from land?

P13/L25-26 – Why just photodegradation without mentioning respiration. Both are important.

P14/L1-2 – Cite appropriate references here (Johnston et al. 2020; also Bogard et al. 2019 Nature Geoscience). Also, as discussed in those papers, elevated DOC in many lakes is due to intense autotrophic inputs.

P14/L5 – Elevation and catchment slope would be an easy factor to add. See general comments. This exploration would boost the mechanistic insight in the paper.

P14/L12 – Some studies in the reference list have done this and could be explored.

Some useful references:

Osburn, C.L., Anderson, N.J., Leng, M.J., Barry, C.D. and Whiteford, E.J., 2019. Stable

isotopes reveal independent carbon pools across an Arctic hydro‐climatic gradient: Implications for the fate of carbon in warmer and drier conditions. Limnology and Oceanography Letters, 4(6), pp.205-213.

Tank, Suzanne E., Lance FW Lesack, Jolie AL Gareis, Christopher L. Osburn, and Ray H. Hesslein. Multiple tracers demonstrate distinct sources of dissolved organic matter to lakes of the Mackenzie Delta, western Canadian Arctic. Limnology and Oceanography 56, no. 4 (2011): 1297-1309.

Bogard, M.J., Kuhn, C.D., Johnston, S.E., Striegl, R.G., Holtgrieve, G.W., Dornblaser, M.M., Spencer, R.G., Wickland, K.P. and Butman, D.E., 2019. Negligible cycling of terrestrial carbon in many lakes of the arid circumpolar landscape. Nature Geoscience, 12(3), pp.180-185.

---

## Author Comment (AC1) · 5 Feb 2021

*Detailed review replies of*

**First Pan-Arctic Assessment of Dissolved Organic Carbon in Permafrost-Region Lakes**

Lydia Stolpmann[1,2], Caroline Coch[1,2,3], Anne Morgenstern[1], Julia Boike[1,4], Michael Fritz[1], Ulrike Herzschuh[1,2,5], Kathleen Stoof-Leichsenring[1], Yury Dvornikov[6], Birgit Heim[1], Josefine Lenz[1,7], Amy Larsen[8], Katey Walter Anthony[7], Benjamin Jones[9], Karen Frey[10], and Guido Grosse[1,2]

[1]Alfred Wegener Institute, Helmholtz Centre for Polar and Marine Research, Potsdam, Germany
[2]Institute of Geosciences, University of Potsdam, Potsdam, Germany
[3]World Wildlife Fund, The Living Planet Centre, Rufford House, Brewery Road, Working, Surrey, GU214LL
[4]Geography Department, Humboldt-Universität zu Berlin, Germany
[5]Institute of Biochemistry and Biology, University of Potsdam, Potsdam, Germany
[6]Agrarian-Technological Institute, Peoples' Friendship University of Russia, Moscow, 117198, Russia
[7]Water and Environmental Research Center, University of Alaska, Fairbanks, AK, USA 99775
[8]Yukon-Charley Rivers National Preserve and Gates of the Arctic National Park and Preserve, National Park Service, Fairbanks, AK, USA
[9]Institute of Northern Engineering, University of Alaska, Fairbanks, AK, USA
[10]Graduate School of Geography, Clark University, Worcester, MA, USA

*Correspondence to*: Lydia Stolpmann (Lydia.Stolpmann@awi.de)

*The authors have presented a synthesis of a very large data set of DOC concentrations from permafrost affected lakes spanning the entire Arctic and a straight forward, but yet effective regional scale analysis of the landscape controls on the DOC concentrations in these lake.*

*The authors have identified the data gap and clearly spelled out the objectives of the study. They have done a commendable job at gathering this very impressive and novel data set, they have provided a detailed summary of the temporal and spatial nature of the data, and have applied valid statistical tools to investigate and identify the role that a range of landscape variables play in controlling the DOC concentrations in Arctic/permafrost lakes. The authors have also done a very good job at addressing the uncertainties and challenges with the data in the discussion.*

Thank you very much for acknowledging the importance of the data presented in our paper. We are grateful for the review and acknowledge the reviewer's comments, which improved and strengthened the paper.

*On it's own the act of gathering and describing this data set constitutes a valuable contribution, and the analyses conducted and conclusions drawn add significantly the value of that contribution. The research presented is well within the scope of the journal and should be of interest to a wide audience. As such I believe the work is definetly worthy of publication. I do feel that the data have been under-utilized in a few respects, and/or there are some analyses that could be added to address some of the issues of uncertainty and the variability in the data and strengthen the paper. I detail the suggestions in the specific comments below. I view these as minor to moderate revisions, as they require some additional analyses or description of the data, but I do not believe they will change the outcomes or conclusions that are reached.*

Thank you very much for the appreciation of our dataset. The dataset we presented here will be basis for further analyses and we intend to use it for further detailed investigations.

Specific Comments:

*Title – I suggest a couple minor changes. First I would remove "-Region" from the title - simply state Permafrost Lakes. Second I would advise the authors to consider editing the title slightly to specify the nature of the "Assessment" conducted. Perhaps: "First Pan-Arctic Assessment of landscape characteristics as controls on Dissolved organic carbon in Permafrost Lakes" Or "First Pan-Arctic Assessment of environmental parameters as drivers for Dissolved organic carbon concentrations in Permafrost Lakes" Or along these lines.*

We like the idea of specifying the title and thank you very much for concrete suggestions. Since the term "permafrost lakes" might be confusing to other readers with respect to the well-known "glacial lakes" or "thermokarst lakes" in the existing literature we adjusted the title accordingly to "First Pan-Arctic Assessment of Dissolved Organic Carbon in Lakes of the Permafrost Region".

*Study Area: The authors spend several paragraphs on pages 3-5 describing the percentages of the lakes that are from the different Arctic regions and the different ecozones etc. Although the authors do provide the lake/sample numbers as n values in Table 2 and Figure 3, because there is a strong spatial bias (with more than half the lakes in Alaska) I suggest that a histogram be included, to better illustrate the geographical distribution of the lakes. This could be included as an insert for example in Figure 1 (using this regional description).*

We agree that the overview of the lakes in the several study areas with n-values in figures and tables and percentages within the text can be confusing. The suggested histogram is indeed a

nice way to better illustrate the lake geographical distribution in our synthesis dataset. We included such a histogram in figure 1 as subfigure 1b) and 1c) see below.

[Figure]

**Figure 1: Overview of study regions (underlined bold font), study areas (bold font) and sites overlain of the map of permafrost zones (a), and histogram of the number of lakes in percentage by study area in our synthesis dataset (b) and pie chart of lake distribution in the dataset by overarching regions (c) (Background map: after Brown et al., 1997).**

*The other aspect of the data that is not clearly illustrated is the timeframe of collection. It would help the reader understand the data, if there was some illustration of the number of samples/lakes that were taken from the various years (e.g. how many samples are from 1979-1985 in Nunavut?). I do not think it would require much effort to generate a temporal histogram (with the number of samples from the various years and quarters).*

Indeed, the number of samples from the various year is very informative for the reader. So, we added the number of samples in brackets after each year in Table 1. Additionally, we added a histogram of numbers of samples per decade per study area to the Appendix as Figure A1, see below. The figure illustrate that there are only a few samples collected in the 90s and two third of the samples were collected since 2009. More information can be found in the dataset itself, which we will publish in the open access data archive PANGAEA soon (doi will be provided with this paper).

[Figure]

**Figure A1: Histogram of numbers of samples in each study area per decade of sample collection. Since an exact allocation of the sampling year was not possible for 113 samples, these are missing in this figure for study areas Nunavut and Manitoba, Canada. The study area Manitoba, Canada, is not included.**

*The authors need to address the potential impact of including samples from nearly 40yr span of time could have on the analyses, especially given that climate and permafrost has been changing dramatically (and at different rates) across these ecoregions during this time. For example this long time frame of the sample period could play a role in the relationship between DOC and latitude. Is it possible the relationships might be more robust if the authors limited themselves to data from the last decade or two? The temporal histogram suggested above would provide a means to speak to this issue.*

Thank you very much for this constructive comment. We included this topic in the discussion. In the past 40 years, the study areas may of course have changed due to the accelerating climate change. Thermokarst lakes in particular are very active. Some lakes that were sampled 30-40 years ago may by now be completely drained and thus no longer exist, for others environmental characteristics in catchments such as permafrost, vegetation cover, or runoff dynamics may have changed over time thereby also affecting lake DOC. The goal of our first synthesis step was not to quantify change in DOC over time but to look a broad spatial patterns on a hemispheric scale. In addition, our dataset shows that more than two thirds of the samples were collected since 1999 (see Table 1, which was modified after the reviewer's comments, and the histogram added to the appendix Figure A1). Hence, analyses with data from the last two decades already form the majority (89 %) of our data set. We agree, however, that continued efforts to expand the dataset in the future should certainly look into the temporal dynamics of northern lake DOC as well.

*Results: Temporal Variability of the data set. I like that the authors indicate the nature of the seasonality of the sampling. Althought section 4.1 breaksdown how many lakes were sampled multiple times, and how the DOC concentrations varied in terms of the overal range and trends in concentrations of DOC (i.e. across all lakes) through different seasons, I feel what is lacking is an assessment of the degree to which DOC concentrations vary in individual lakes over the year or seasons. i.e. is the range of variability within a year/season in a given lake greater than the variability between lakes for this subset? I suspect this seasonal variability is minimal, however if seasonality is important then the authors would need to consider limiting the data set to lakes sample from a consistent time of the year? Although the temporal subset is small relative to the whole data set, there are 81 lakes and 266 samples, which is by most any measure still a substantial data set. There is likely sufficient data within this to provide some assessment of the relative impact of both the seasonality on the data.*

We agree that the DOC seasonality of our dataset provides an opportunity to assess the degree to which DOC concentrations vary in individual lakes over the year or season. We now address this in the results and discussion. We added Table A2 to the Appendix and the following to subchapter 4.1:

"In our dataset we found six lakes that were sampled three times over the same period. These six lakes are located in Qeqqata on Greenland and were sampled in April, June and August. For five of these six lakes, the highest DOC concentration of the respective sampling series was found for April samples. Then, the DOC concentration decreased in June and increased in August (Tab. 2). For these lakes, we observed a 30 % to 45 % higher DOC concentration in April and up to 25 % higher DOC concentration in August in comparison to the June sampling and therefore shows a seasonal variability. We have also checked for seasonal variability in a larger dataset. Therefore, we used data from the study areas Southcentral and Interior Alaska since samples in these areas were available for each month from May to September. However, these samples include all samples of these study areas and not a sample series of a single lake. We focused on the median DOC concentration for each month for each of the two study areas. For Southcentral Alaska we found a pattern similar to that in the lakes in Qeqqata. We found a 17% higher DOC concentration in May and September compared to July. However, the same pattern cannot be seen for Interior Alaska (Table A2).
In order to make a more precise comparison with the Qeqqata lakes, we compared samples of the whole dataset from the months June and August. For these months, in addition to the Qeqqata samples, samples from the study areas Yamalo-Nenets A.R., North and Northwest Alaska, Southcentral and Interior Alaska were available. In three of the four study areas we also found higher DOC concentrations in August than in June, comparable to the Qeqqata lakes."

**Table 1: DOC concentration of six lakes (Qeqqata, Greenland) sampled three times in one year.**

| Lake name | DOC concentration [mg L$^{-1}$] | | |
|---|---|---|---|
| | **April** | **June** | **August** |
| SS906 | 8 | 5.2 | 5.8 |
| SS1381 | 39.2 | 24 | 31.3 |
| SS2 | 35.1 | 23.8 | 27.7 |
| SS8 | 52.5 | 28.7 | 38.3 |
| SS904 | 8.1 | 5.2 | 5.8 |
| SS1590 | 31.1 | 36.6 | 25 |

**Table A2: Overview of median DOC concentration according to sampling months, where sampling month could be clearly identified.**

| | Study area | Median DOC concentration [mg L$^{-1}$] | | | | | | |
|---|---|---|---|---|---|---|---|---|
| | | **April** | **May** | **June** | **July** | **August** | **September** | **October** |
| **Greenland** | Qeqqata | 33.1 | | 10.2 | | 25 | | |
| **Siberia** | Yamalo-Nenets Autonomous Region | | | 18.1 | 29.6 | 13.9 | | |
| | Khanty-Mansi Autonomous Region | | | | 14.1 | 10.9 | | |
| | Chukotka Autonomous Region | | | | 9.6 | | | |
| | Krasnoyarsk Krai | | | | 8.3 | 9.8 | | |
| | Sakha Republic (Yakutia) | | | | 8.1 | 10.2 | 7.8 | 15.3 |
| **Canada** | Yukon | | | | 12.9 | | | |
| | Northwest Territories | | | | 8.5 | | 13 | |
| | Nunavut | | | | 2.2 | 3.8 | 2.6 | |
| **Alaska** | North and Northwest | | | 4.5 | 9.1 | 8.2 | 8.1 | |
| | Southcentral | | 15 | 13.6 | 12.5 | 16.3 | 15 | |
| | Interior | | 12.7 | 18.4 | 12.6 | 20.3 | 21.3 | |

To clarify, that we didn't use under-ice samples for our statistical analysis but here to investigate seasonality of our data, we added the following sentence to subchapter 3.3 Statistical analysis:

"Our dataset contains six samples from Qeqqata on Greenland that were collected in April and are under-ice samples. For the sake of comparability, these data have not been included in the statistical analysis."

We also added this point to the discussion subchapter 5.4 Challenges of a pan-Arctic assessment, describing possible causes of the seasonal variability found in a small scale single lake analysis and similar patterns analyzing multiple lakes per month. Additionally, we highlight the lack of seasonal data to emphasize that seasonal variability and that continuous sampling during the ice-free period is needed.

*I didn't see a data availability statement. Even if this is not a requirement of the journal, it is critical and needs to be included – perhaps in the results or cited as a data archive? Readers are going to want to know how or where they might be able to access this valuable data set.*

Thank you for the comment. We are currently preparing the dataset for publication in PANGAEA and will provide the DOI soon.

Technical corrections:

*P1 L29 Edit: Our synthesis shows a significant relationship of lake DOC concentration and ecoregion of the lake. \*cut "of" insert "between" , insert "lake" ahead of "ecoregion" and cut "of the lake"*

We changed this accordingly.

*P1L32 Compared to previous studies we found a weak significant relationship of soil organic carbon content. . .. \*cut "of" insert "between"*

We changed this accordingly.

*P7 Line 21 - add "data" to the end of the section title*

We changed this accordingly.

*P13 L16 change "with a surface are " to ". . .surface area"*

We changed this accordingly.

*P.21 L20 – check reference volume number "Limnol. Oceanogr., 9999"?*

Thank you very much. We corrected it into: Johnston, S. E., Striegl, R. G., Bogard, M. J., Dornblaser, M. M., Butman, D. E., Kellerman, A. M., Wickland, K. P., Podgorski, D. C., and

Spencer, R. G. M.: Hydrologic connectivity determines dissolved organic matter biogeochemistry in northern high-latitude lakes, Limnol. Oceanogr., 65(8), 1764-1780, doi: 10.1002/lno.11417, 2020.

---

## Author Comment (AC2) · 5 Feb 2021

*Detailed review replies of*

**First Pan-Arctic Assessment of Dissolved Organic Carbon in Permafrost-Region Lakes**

Lydia Stolpmann[1,2], Caroline Coch[1,2,3], Anne Morgenstern[1], Julia Boike[1,4], Michael Fritz[1], Ulrike Herzschuh[1,2,5], Kathleen Stoof-Leichsenring[1], Yury Dvornikov[6], Birgit Heim[1], Josefine Lenz[1,7], Amy Larsen[8], Katey Walter Anthony[7], Benjamin Jones[9], Karen Frey[10], and Guido Grosse[1,2]

[1]Alfred Wegener Institute, Helmholtz Centre for Polar and Marine Research, Potsdam, Germany
[2]Institute of Geosciences, University of Potsdam, Potsdam, Germany
[3]World Wildlife Fund, The Living Planet Centre, Rufford House, Brewery Road, Working, Surrey, GU214LL
[4]Geography Department, Humboldt-Universität zu Berlin, Germany
[5]Institute of Biochemistry and Biology, University of Potsdam, Potsdam, Germany
[6]Agrarian-Technological Institute, Peoples' Friendship University of Russia, Moscow, 117198, Russia
[7]Water and Environmental Research Center, University of Alaska, Fairbanks, AK, USA 99775
[8]Yukon-Charley Rivers National Preserve and Gates of the Arctic National Park and Preserve, National Park Service, Fairbanks, AK, USA
[9]Institute of Northern Engineering, University of Alaska, Fairbanks, AK, USA
[10]Graduate School of Geography, Clark University, Worcester, MA, USA

*Correspondence to*: Lydia Stolpmann (Lydia.Stolpmann@awi.de)

*The manuscript presents a newly assembled, Global dataset of DOC concentrations from northern lakes. It is a very nice dataset that is clearly valuable. The mapping of patterns of DOC throughout figures 1 to 3 and tables 1 to 2 are also very useful context for readers. On this basis, I feel the manuscript has great potential to make an important contribution to the field. However, while the dataset is powerful, and geographic patterns are interesting, the paper is lacking in mechanistic insight, with a number of important concepts overlooked. These issues should be resolved before the manuscript could be considered fit for publication. I outline my concerns below, and hope that the authors find this evaluation constructive and useful.*

Thank you very much for the appreciation of our dataset and the constructive comments.

General comments:

*1. I think that the core message of the paper needs to change from one of strongly predicting lake DOC, to stating that predicting DOC patterns at the Global scale is complex, and has resulted in weak*

*relationships with the individual predictors at hand. Throughout, the authors claim that lake DOC strongly depends on environmental properties. I agree, yet I don't think that the results presented here have led to this conclusion. Disregarding p values (which are uninformative due to their dependence on sample size), the correlation strength for every environmental parameter versus DOC is weak, and weaker than those for basic categorical groupings (region/zone) and latitude, which themselves are marginal in strength. The fact that these categorical and spatial variables remain stronger predictors than environmental measures (ice content, ground type, soil C content) tells us that the major drivers of lake DOC are not captured in this dataset (see next point on which predictors I mean). That isn't surprising, and speaks to the complex regulation of lake DOC concentrations. To fix this problem, I think that the messaging of the paper needs to change to emphasize the weakness of the strength of these individual predictors, and the complex control of lake DOC content has to be discussed throughout and emphasized. Throughout, I feel that readers are presented with an over-simplified view of the regulation of DOC concentrations.*

We would like to emphasize that our objectives were to synthesize existing lake DOC data to provide a pan-Arctic dataset, and to present what we conclude from this dataset. Our synthesis shows how important local studies of single lake catchment are to explain the complex mechanisms of regulating lake DOC. This dataset should be basis for further investigations. We agree that the correlations for our environmental parameters and the lake DOC concentration are weak but we wanted to highlight the strongest relationships in our dataset, even if these are rather weak. However, we also agree that it is necessary to highlight that these relationships are weak and that we need to describe and discuss the complexity of regulating DOC concentration in lakes more in detail in the discussion. Therefore we added an additional subsection to the discussion, we attached here:

**"5.3 The complexity of lake DOC regulation**

The analysis with our dataset with the available pan-Arctic data have shown the strongest significant relationships between ecoregion and lake DOC concentration as well as between geographical latitude and DOC concentration and permafrost extent and DOC concentration, even if these relationships are generally weak. Recent studies present more parameters influencing lake DOC concentration. Xenopoulos et al. (2003) analysed catchment characteristics of lakes and found that lake perimeter and the proportion of watershed occupied wetlands are strongest predictors for DOC in lakes of the temperate forest. On a global scale, the proportion of watershed occupied wetlands and lake elevation are strongest predictors for lake DOC. Tranvik et al. (2009) described that lake are, which is connected to lake volume and water retention time, might be negatively correlated in regional studies but that it is not an important DOC predictor in a global view. The fact that the majority of predictors for lake DOC differ in regions demonstrate the complexity of the regulation of DOC concentration in lakes. However, these parameter are not included in our study, which could cause the weak relationships of our analysis. Also not included in our analysis due to limited data availability is the hydrological connectivity. Therefore, less allochthonous (i.e. produced in the lake catchment) DOC is transported to hydrologically isolated lakes (Bogard et al., 2019). Instead,

those lakes are characterized by autotrophic C inputs by fixing atmospheric $CO_2$ (Bogard et al., 2019). In these lakes, such as in the Yukon Flats, also evapoconcentration plays an important role (Johnston et al., 2020). In addition to allochthonous DOC, autochthonous (i.e. produced in the lake) DOC is influencing the DOC concentration, especially in lakes with low connectivity. Autochthonous DOC includes phytoplankton productivity as well as heterotrophic bacterioplankton respiration processes (Chupakov et al., 2017).

The influence of hydrological, climatic and topographical parameter on the DOC concentration of a lake is very complex. Whereas our pan-Arctic dataset provides first insights of the relationship between environmental parameter and lake DOC concentration, regional studies are necessary to understand these complex mechanisms and to determine DOC predictors, which are differing regionally. In addition to the described terrestrial DOC predictors, newest"

*2. The paper overlooks the most important mechanisms controlling DOC cycling, i.e., the roles of hydrology and geomorphology in structuring the delivery of DOC to lakes, and water residence time and allochthonous DOC processing. This may be a main reason for such weak relationships presented in their correlation table. Many other studies show that these factors critically shape lake DOC cycling, and without this information brought in to the analyses, the authors are likely missing a big part of the mechanistic story. For instance, the first paragraph of the discussion talks about vegetation density, soil C contact with water, and the effects of permafrost extent on vegetation. It really overlooks these major factors that are very much a part of that story, thereby making the mechanistic explanations in the paper incomplete. To fix this, the authors could improve the analyses by bringing in new datasets on hydrology, topography (catchment slope), and estimates of water residence time (even for a subset of the lakes) to explore this. At the very least, the importance of these factors needs to be better incorporated into the narrative of the paper, from start to finish.*

We agree that we need to clarify that important aspects for the regulation of the DOC concentration, such as hydrology and topography, were not included in our pan-Arctic analysis and that this has an influence on the results. Recent studies on lake size distribution were inconclusive (Muster et al., 2019) and statistical analysis showed that the lake size distribution could not be explained by hydrological variables such as TDD s, evapotranspiration, precipitation, permafrost extent, glaciation history, or broad-scale topographic parameters (i.e., elevation and slope). At the moment, we cannot subject this aspect to an additional analysis, as this would go beyond the scope of the current project possibilities. But we included the description of the role of these important factors in our paper, especially in the discussion subsection 5.3 (see above), and incorporated them into the conclusions as an outlook for further exciting work.

*3. The paper overlooks the role of autochthonous DOC production, and its importance in structuring patterns of DOC within and among regions. It is clear from past work using a bunch of different approaches in different regions, that the sources and composition of DOC ranges widely, and autochthonous sources can be very important in many lakes (e.g., Tank et al. 2011 L&O; Osburn et al. 2017 JGR-B; Osburn et al. 2019 L&O Lett.; Johnston et al. 2020 L&O). A lot of the high-DOC lakes in*

*some regions are indeed rich in autochthonous DOC, so this source input may drive some of the most extreme observations in the current dataset, especially in the Yukon Flats region the authors highlight, which is discussed in that Johnston et al. 2020 paper. The importance is probably lower in some regions than others, and depends on hydrologic connections to the terrestrial landscape and other factors (getting back to comment 2 above). Overall, this is an important factor structuring lake DOC patterns that should at the very least be incorporated throughout the paper, where mechanistic inferences are made. To go a step further, the authors could look at a subset of the lakes for which published organic matter properties are available (optical, isotopic, elemental ratios, or other, depending on what is out there).*

We agree that the role of autochthonous DOC production should also be examined and discussed, which we have added in our paper at subsection 5.3 (see above).

Specific comments:

*P1/L32 – How does this relationship compare, specifically?*

In this sentence we refer to our results regarding analyzing the relationship between lake DOC concentration and soil organic carbon content. In the results subsection 4.7 we described that our analysis result in a weak significant relationship between lake DOC and SOCC of the upper 100 cm ($\rho = 0.1$; $p < 0.05$). Analysing the relationship of lake DOC and SOCC in the upper 300 cm we found a weaker significant relationship ($\rho = 0.09$; $p < 0.05$). In comparison, Sobek et al. (2007) assumed soil carbon content as a strong predictor for lake DOC.

*P1/L34 – I don't think you demonstrate a strong dependence here, see general comment 1.*

We deleted 'strongly' in the sentence.

*P2/L4 – abbreviate carbon as C after first mention. Comment applies throughout.*

We changed it accordingly.

*P2/L11 – regional warming: is this air or soil temperature? Be specific.*

We specified in the manuscript that there was found a permafrost warming driven by higher air temperatures:

"More recently, the permafrost warmed globally by an average of 0.29° +/- 0.12° C over the 2007-2016 period due to higher air temperatures, with some of the strongest warming trend (about 0.9° C per decade) measured in individual boreholes in Siberia (Biskaborn et al., 2019)."

*P2/L28 – Which lake-based process, specifically?*

Here we mean the mineralization by photo oxidation or microbial activity, resulting in $CO_2$ and $CH_4$ emission. We clarify and changed the sentence to:

"The mineralization of DOC in a lake is a major component of the global carbon cycle and contributes to the greenhouse effect (Finlay et al., 2006). Vonk et al. (2015) suggested that this

form of carbon flux represents roughly one third to one-half of the net carbon exchange from land to the atmosphere in the Arctic."

*P2/L30 – Last sentence needs improving: Which flux? Lakes, streams, what? The paragraph could also be improved by adding a conclusion sentence identifying what the major knowledge gap is here.*

In this paragraph we want to highlight the importance of analyzing DOC in lakes because lake-based process, such as mineralization by photo oxidation or microbial activity, are a major component of the global carbon cycle and contributes to the greenhouse effect (Finlay et al., 2006). To clarify the understanding of the sentence we revised it:

"Vonk et al. (2015) suggested that this form of carbon flux, comprising the carbon flux from surface waters to the atmosphere and from land to ocean, represents roughly one third to one-half of the net carbon exchange from land to the atmosphere in the Arctic."

*P3/L1-9 – This paragraph is somewhat off topic. What does it have to do with lake DOC? Either revise it to link to the topic, or cut it out.*

Thank you for this remark. We agree that this sentence about riverine DOC does not fit here and decided to cut the sentence out.

*P3/L18 – Lake number or lake area?*

Here we mean 10 % of the lake number of 7,500. To clarify that we also here refering to the study of Sobek et al. (2007), we added the reference to the end of the sentence.

*P3/L25 – Throughout: Cold/Very Cold/Cool – give us some meteorological values to help understand what these designations mean.*

Thank you for this comment, which makes clear that the constant repetition of 'cold/very cold/cool' does not highlight climatic differences between the study areas. We revised the section by cutting the climate classifications in each study area paragraph and by including the climate classification at the beginning of the section as follows:

"In our synthesis, we included 2,167 samples from 1,833 lakes of 13 study areas (22 sites) across the Arctic, sampled from year 1979 to 2017 (Table 1). The study areas of our dataset are dominated by tundra climate and very cold subarctic climate. The Nunavut study area is also characterized by cool continental climate."

*P7/L21 – This section provides little insight. As it is a paper on global patterns, the seasonality seems off topic. What's more, there is no context to evaluate DOC patterns in a meaningful way (which lakes, which regions, etc.). Consider removing this.*

We agree that the section provides only little insights. This was also criticized by the first reviewer. For this reason we have expanded this section. Enclose you will find the additions we made. We added Table A2 to the Appendix and the following to subchapter 4.1:

"In our dataset we found six lakes that were sampled three times over the same period. These six lakes are located in Qeqqata on Greenland and were sampled in April, June and August. For five of these six lakes, the highest DOC concentration of the respective sampling series was found for April samples. Then, the DOC concentration decreased until sampling in June and increased until sampling in (Tab. 2). For these lakes, we observed a 30 % to 45 % higher DOC concentration in April and up to 25 % higher DOC concentration in August in comparison to the June sampling and therefore shows a seasonal variability. This variability we checked for a larger dataset. Therefore, we used data from the study areas Southcentral and Interior Alaska since samples in these areas were available for each month from May to September. However, these samples include all samples of these study areas and not a sample series of a single lake. We focused on the median DOC concentration for each month for each of the two study areas. For Southcentral Alaska we found a pattern similar to that in the lakes in Qeqqata. We found a 17% higher DOC concentration in lakes that were sampled in May and September compared to lakes that were sampled in July. However, the same pattern cannot be seen for Interior Alaska (Table A2).

In order to make a more precise comparison with the Qeqqata lakes, we compared samples of the whole dataset from the months June and August. For these months, in addition to the Qeqqata samples, samples from the study areas Yamalo-Nenets A.R., North and Northwest Alaska, Southcentral and Interior Alaska were available. In three of the four study areas we also found higher DOC concentrations in August compared to the June samples like for the Qeqqata lakes."

**Table 1: DOC concentration of six lakes (Qeqqata, Greenland) sampled three times in one year.**

| Lake name | DOC concentration [mg L$^{-1}$] | | |
|---|---|---|---|
| | April | June | August |
| SS906 | 8 | 5.2 | 5.8 |
| SS1381 | 39.2 | 24 | 31.3 |
| SS2 | 35.1 | 23.8 | 27.7 |
| SS8 | 52.5 | 28.7 | 38.3 |
| SS904 | 8.1 | 5.2 | 5.8 |
| SS1590 | 31.1 | 36.6 | 25 |

**Table A2: Overview of median DOC concentration for sampling months. Includes those samples that could be assigned to a sampling month unambiguously.**

| Study area | | Median DOC concentration [mg L$^{-1}$] | | | | | | |
|---|---|---|---|---|---|---|---|---|
| | | **April** | **May** | **June** | **July** | **August** | **September** | **October** |
| **Greenland** | Qeqqata | 33.1 | | 10.2 | | 25 | | |
| **Siberia** | Yamalo-Nenets Autonomous Region | | | 18.1 | 29.6 | 13.9 | | |
| | Khanty-Mansi Autonomous Region | | | | 14.1 | 10.9 | | |
| | Chukotka Autonomous Region | | | | 9.6 | | | |
| | Krasnoyarsk Krai | | | | 8.3 | 9.8 | | |
| | Sakha Republic (Yakutia) | | | | 8.1 | 10.2 | 7.8 | 15.3 |
| **Canada** | Yukon | | | | 12.9 | | | |
| | Northwest Territories | | | | 8.5 | | 13 | |
| | Nunavut | | | | 2.2 | 3.8 | 2.6 | |
| **Alaska** | North and Northwest | | | 4.5 | 9.1 | 8.2 | 8.1 | |
| | Southcentral | | 15 | 13.6 | 12.5 | 16.3 | 15 | |
| | Interior | | 12.7 | 18.4 | 12.6 | 20.3 | 21.3 | |

We also added this point to the discussion subchapter 5.4 Challenges of a pan-Arctic assessment, describing possible causes of the seasonal variability found in a small scale single lake analysis and similar patterns analyzing multiple lakes per month. Additionally, we name the lack of data to make a more solid statement about seasonal variability and that continuous sampling during the ice-free period is needed.

*P8/L7 – Where is table A2? Why not cite Fig. 2 & 3 with statistics reported to confirm this statement. Comment applies throughout paper about table A2.*

Table A2 contains the exact DOC concentration ranges, median DOC concentration values and mean values for each individual study site in the various study areas. However, since this table is very large, we decided to move it to the appendix. In this sentence, we refer to table A2 because it provides exact values as differences between the sites. However, we agree that the different DOC concentrations in the study sites are also very well demonstrated in Figures 2 and 3 and that we should also refer to them here. That's why we added:

"We found differences between the four regions of Alaska, Canada, Greenland and Siberia, as well as between study areas and study sites within these regions (Fig. 2, Fig. 3, Table A2)."

*P10/L1 – Instead of fig. 2a, why not cite Fig 3 here? Makes the point better.*

We agree that Figure 3 highlights the variability of median DOC concentration in the study regions on a better way. We changed it accordingly.

*P10/L5-6 – Why a new paragraph? Same topic/theme.*

Thank you very much for this comment. We changed it accordingly.

*P11 – Fig. 3 – Where are the statistics? ANOVA and post hoc tests for each panel?*

For our statistical analysis we first used the Shapiro–Wilk normality for testing whether our data is normality distributed. But our data does not follow a normal distribution. So, we did not use the ANOVA but the Spearman rank correlation coefficient (ρ) to measure the relationship between two variables and the Wilcoxon-Mann-Whitney test to determine the difference in means between two populations. We described this on page 7 in the subsection 3.3 Statistical Analysis.

*P12/L14-16 – Adding scatterplots and show us the data – the table is good but not enough.*

We added the following scatterplot to this paragraph to better visualize the data of lake DOC concentration and soil organic carbon content.

Additionally, we would like to notice that the entire dataset is in progress for publication in PANGAEA.

[Figure]

**Figure 4: Scatterplots for lake DOC concentration and lake surrounding soil organic carbon content (SOCC) in a depth of 0 to 100 cm (a) and in a depth of 0 to 300 cm (b). To better visualize the relationship of both parameter we limited the y-axis to 200 mg L$^{-1}$. So, three lakes with the DOC concentrations of 433 mg L$^{-1}$, 507 mg L$^{-1}$ and 1,130 mg L$^{-1}$are not included in this plot.**

*P12/L24 – Does the study provide this insight? I don't think so, since the relationships are quite weak. See general comments for suggestions to improve this section.*

We agree that we had to improve the insights of potential sources of DOC in pan-Arctic lakes of the permafrost region. With the help of the reviewers comment we were able to strengthen the manuscript and to better provide this insights, especially by extending the discussion section. Please find the improvements in the following comments and in the response to the first general comment.

*P13/L4 – Should the word 'catchment' be plural?*

Thank you very much for this comment. We changed it accordingly.

*P13/L7 – What about the effect on hydrologic connectivity caused by permafrost extent? Not just terrestrial veg distribution, but also consider the role of permafrost in disconnecting lake surface waters from hydrologic flowpaths that deliver DOC.*

We agree that hydrological and carbon cycles are closely linked, which has been stated in previous publications (Vonk et al., 2015; Beel et al., 2020). We did not analyse the hydrological connectivity (both vertical and lateral) in this paper, but added the importance of the hydrology in the discussion. Additionally, we agree to mention the hydrological connectivity here and extended the sentence.

"In contrast, higher permafrost extent in higher geographical latitudes results in lower vegetation density, lakes are less connected and thereby hydrologically isolated leading to lower DOC concentrations".

*P13/L9 – Odd conclusion sentence. Seems unrelated. Consider revising.*

We agree that this sentence about riverine DOC sources is unrelated in this paragraph about the relationship between lake DOC and ecoregion. Therefore, we decided to cut the sentence out.

*P13/L12 – Weak cross-regional correlations don't really tell us about climate change impacts. Consider revising.*

We revised this part to clarify our point:

"Climate change affects ecosystem structure which in turn may impact lake DOC concentrations and associated biogeochemical fluxes (Sobek et al., 2005). For example, enhanced DOC concentrations in a lake provide an increased basis for the mineralization of DOC through photo oxidation and by microbial activities, which may result in higher $CO_2$ emissions from these lakes. In our first pan-Arctic assessment of DOC in lakes of the permafrost region we found that DOC concentrations in lakes become significantly higher when transitioning from tundra zone to the tundra-boreal transition zone to the boreal zone. In addition, DOC concentrations are overall lower in permafrost zones that are less continuous. Both trends suggest that climate change, projected to result in expansion of boreal zone into the tundra zone and a decrease in permafrost continuity will likely result in higher DOC concentrations in lakes of these regions. Moreover, the shift of the boreal forest might lead to changes in the hydrological connectivity to more connected lakes and further might lead to an

increase of allochthonous DOC in lakes. This may then also result in higher $CO_2$ fluxes from pan-Arctic lakes."

*P13/L13 – The discussion throughout this last paragraph assumes all DOC comes from terrestrial environments. Not accurate. See general comment #3.*

We agree that not all DOC comes from terrestrial environments. In addition to the new subsection 5.3 where we discussed the complexity of lake DOC regulation, we revised the paragraph you find above at the previous comment.

*P13/L20 – "Our results" – be specific here and say which results.*

We specified and changed it accordingly.

*P13/L20 to 21 – You do not have the data to infer this mechanism. Revise language here. Also, I do not think that this conclusion is consistent with the comparison of DOC concentrations by deposit type categories in Fig. 3. Yedoma type is not the highest DOC concentration there.*

It is right that we first considered the DOC concentrations in all occurring deposit types of our dataset. Here we found the highest DOC concentrations in lakes surrounded by mountain alluvium, glacio-lacustrine and glacial deposits and not in lakes surrounded by yedoma deposits (Fig. 3). Then, we focused on yedoma lakes and non-yedoma lakes, including all lakes that were not assigned to yedoma deposits. For yedoma lakes we found a median DOC concentration of 11.8 mg $L^{-1}$, comprising 16 % of the datasets lakes. This is significantly higher than the median DOC concentration we found for non-yedoma lakes with 10.3 mg $L^{-1}$, comprising 79 % of the datasets lakes (section 4.5 of the manuscript). To clarify our point we revised this part:

"The comparison of the DOC concentration in lakes in the yedoma region and the DOC concentration in lakes in non-yedoma regions, comprising all other lakes of our dataset, results in significantly higher DOC concentrations in yedoma lakes compared to non-yedoma lakes. This might be attributed to the mobilization of old labile yedoma carbon by thermos-erosion along rapidly expanding lake shores and thermokarst processes, which (Strauss et al., 2017). Nevertheless, the partial dataset of non-yedoma lakes comprises 79% of lakes in our dataset, including lakes surrounded by different deposit types and lakes of highest DOC concentration in our dataset."

*P13/L23-24 – You can't assume this without hydrologic or other information.*

*P13/L25 – This could be revised to be more mechanistically insightful. Wouldn't we expect lower delivery of allochthonous DOC to these lakes where permafrost limits hydrologic exchange into lakes from land?*

*P13/L25-26 – Why just photodegradation without mentioning respiration. Both are important.*

We agree with the last three comments that in this paragraph more explanation and discussion is needed and we decided to revise the sentences on P13/L23 to 26 as follows:

"We assume that yedoma lake generation is influencing yedoma lake DOC. The formation of yedoma lakes, due to deep thermokarst subsidence, results in deep and often closed basins

(Morgenstern et al., 2011). As result of the missing lake connectivity, DOC is locked in the lake, originating partially from eroding organic-rich yedoma deposits (Strauss et al., 2017) and melting yedoma ice wedges (Fritz et al., 2015). Further, the lacking lake connectivity might prevent flushing of yedoma thermokarst lake waters with river water and snowmelt water. Hence, we assume that yedoma thermokarst lakes are more likely to have elevated DOC concentrations than other more connected lakes as well as mixed larger and shallower lakes, where photodegradation and respiration plays an important role, are associated with lower lake DOC concentration."

*P14/L1-2 – Cite appropriate references here (Johnston et al. 2020; also Bogard et al. 2019 Nature Geoscience). Also, as discussed in those papers, elevated DOC in many lakes is due to intense autotrophic inputs.*

We agree that the suggested references need to be mentioned here. We revised as follows:

"For our database, this connection may explain the relatively high DOC concentrations of lakes in the Yukon basin. Here, the lakes are less hydrologically connected and the region is very arid Therefore, evaporation can concentrate DOC (Johnston et al., 2020)".

Additionally, we added the aspect of the autotrophic carbon input described by Bogard et al. (2019) in the discussion section 5.3.

*P14/L5 – Elevation and catchment slope would be an easy factor to add. See general comments. This exploration would boost the mechanistic insight in the paper.*

Thank you very much for suggesting this here but at the moment we cannot subject this aspect to an additional analysis, as this would go beyond the scope of the current project possibilities. We want to mention that our dataset provides a basis for further analysis. We plan to extent the dataset to conduct more analysis and for example use parameter such as elevation or catchment slope.

*P14/L12 – Some studies in the reference list have done this and could be explored.*

We agree that there have been studies carried out in lake areas which are remote in the Arctic. Here we want to highlight that in particular multitemporal sampling of those especially remote lakes is logistically challenging but necessary to investigate the seasonal variability of lake DOC in different regions.

Some useful references:

*Osburn, C.L., Anderson, N.J., Leng, M.J., Barry, C.D. and Whiteford, E.J., 2019. Stable isotopes reveal independent carbon pools across an Arctic hydroăAˇ Rclimatic gradient: Implications for the fate of carbon in warmer and drier conditions. Limnology and Oceanography Letters, 4(6), pp.205-213.*

*Tank, Suzanne E., Lance FW Lesack, Jolie AL Gareis, Christopher L. Osburn, and Ray H. Hesslein. Multiple tracers demonstrate distinct sources of dissolved organic matter to lakes of the Mackenzie Delta, western Canadian Arctic. Limnology and Oceanography 56, no. 4 (2011): 1297-1309.*

Bogard, M.J., Kuhn, C.D., Johnston, S.E., Striegl, R.G., Holtgrieve, G.W., Dornblaser, M.M., Spencer, R.G., Wickland, K.P. and Butman, D.E., 2019. Negligible cycling of terrestrial carbon in many lakes of the arid circumpolar landscape. Nature Geoscience, 12(3), pp.180-185.

Thank you very much for adding this useful publications.

---

## Referee Report (RR1)

A review on a paper
"First Pan-Arctic Assessment of Dissolved Organic Carbon
in Lakes of the Permafrost Region"
by Stolpmann et al.
submitted to *Biogeochemistry*

**General remarks**

The paper presents so far the largest available database on the DOC concentration in Northern permafrost lakes. It will be important contribution to further research of Pan-Arctic carbon budget, which still suffers from significant uncertainties, among which the freshwater ecosystems role is not the least. The manuscript is in a good shape and should be published after a number of comments listed below are addressed. I have two major methodological concerns:

- a potentially important and still poorly quantified source of errors for such global or regional estimates is that the samples from individual sites are taken at different seasons of a year, thus representative of different phases of annual cycle; this makes a month of sampling to be one more factor of DOC concentration in addition to a list of factors studied in the paper; this factor is addressed in Section 4.1 (and it is shown, that for some lakes DOC difference between seasons attains an order of magnitude), but no implications are formulated for analysis carried out in subsequent Sections; in fact, neglecting the difference in season of sampling between lakes imposes uncertainty which is additional to the factors either omitted in this study like climate parameters and local hydrological conditions; I suggest to add analysis of this factor in Discussion;

- in the study, the individual correlations of DOC with different factors are estimated, whereas the *joint* effect of these factors and predictive skill of a *set* of respective parameters on lake DOC content may be estimated as well, concomitantly quantifying the remaining uncertainty imposed by not taking into account the other factors; multiple correlation analysis could be a natural extension of the method used in the paper to achieve such estimates. I suggest that the authors elaborate on this topic in the paper.

**Specific comments**

- Page 2, Line 10: better to specify which is the part of Siberia, where individual boreholes demonstrate significant positive temperature trend
- Page 2, Line 25: is there direct chemical pathway from DOC to $CH_4$? Anyway, this is not mineralization. If you mean that DOC is stored in sediments after flocculation, and then decomposed to $CH_4$, please rephrase the sentence to make it clear.
- Page 2, Lines 26-29: "The mineralization of DOC in a lake is a major component of the global C cycle..." is too strong statement. "one third to one-half" – does this contribution include river C fluxes? Please rephrase to make understandable.
- From which lacustrine layer DOC has been sampled in lakes? I guess, epilimnion. Please provide the info.
- Section 3.3: I would expect, that in order to isolate the influence of a single factor on lake DOC, you should compute correlation coefficient for a series of DOCs from all lakes, for which the other factors are fixed. Or you computed each correlation coefficient for the total number of lakes? Please precise.
- Table 4: not clear, how the rank correlation coefficient have been computed for qualitative predictors: permafrost region, ecoregion, ground ice content, deposit type. Please provide details.
- What is the reason for ultrahigh DOC concentrations in some Alaskan lakes?
- Page 16, Lines 30-31: an unfinished sentence
- Page 17, Line 22: I guess you meant "occupied **by** wetlands"

---

## Author Response (AR2)

General Comments:

*The paper presents so far the largest available database on the DOC concentration in Northern permafrost lakes. It will be important contribution to further research of Pan-Arctic carbon budget, which still suffers from significant uncertainties, among which the freshwater ecosystems role is not the least. The manuscript is in a good shape and should be published after a number of comments listed below are addressed.*

> Thank you very much for acknowledging the importance of the data presented in our paper. We are grateful for the review and acknowledge the reviewer's comments, which improved and strengthened the paper.

*A potentially important and still poorly quantified source of errors for such global or regional estimates is that the samples from individual sites are taken at different seasons of a year, thus representative of different phases of annual cycle; this makes a month of sampling to be one more factor of DOC concentration in addition to a list of factors studied in the paper; this factor is addressed in Section 4.1 (and it is shown, that for some lakes DOC difference between seasons attains an order of magnitude), but no implications are formulated for analysis carried out in subsequent Sections; in fact, neglecting the difference in season of sampling between lakes imposes uncertainty which is additional to the factors either omitted in this study like climate parameters and local hydrological conditions; I suggest to add analysis of this factor in Discussion.*

> We agree that treating all samples from different sampling periods equally leads to further uncertainties that need more attention in our discussion. We added the discussion on differences in DOC concentration over the sampling period in section 5.3 and supplemented to the following paragraph:
>
> "Beside our analysis of temporal variability of a subset of our dataset, the sampling month of each sample was not included in the statistical analysis of our pan-Arctic dataset, which may result in uncertainties due to variations in lake DOC concentration over the ice-free period. For Qeqqata, Greenland, higher DOC concentrations were found in samples collected in April (under ice) and August compared to June samples. In winter, nutrients as well as DOC do concentrate in lakes (Manasypov et al., 2015; Vonk et al., 2015; Grosbois et al., 2017), resulting

in higher DOC concentrations in under-ice samples from April. The spring flood transports large amounts of allochthonous DOC to the lakes, fueling them with DOC and resulting in higher lake DOC concentrations in spring (Manasypov et al., 2015). During summer, low precipitation in this region, evapoconcentration is a major cause for increasing DOC concentration (Anderson & Stedmon, 2007). Considering a seasonality of DOC concentration in our dataset, we found two different patterns in two different study sites. This highlights the complexity of regulators and mechanisms of the DOC concentration in a lake over a season and the need to expand multi-temporal sampling of lake systems."

*In the study, the individual correlations of DOC with different factors are estimated, whereas the joint effect of these factors and predictive skill of a set of respective parameters on lake DOC content may be estimated as well, concomitantly quantifying the remaining uncertainty imposed by not taking into account the other factors; multiple correlation analysis could be a natural extension of the method used in the paper to achieve such estimates. I suggest that the authors elaborate on this topic in the paper.*

Because our data does not follow a normal distribution we decided not to use a multiple correlation analysis. Instead, we performed a principal component analysis (PCA) to estimate the joint effect of DOC and all analysed parameters. Since this effect was not very strong we originally decided to not include the PCA in our manuscript. However, following your comment, we recognized that the PCA is nevertheless of interest and visualizes some correlations in our dataset. We revised section 3.3 (see below) and added the PCA to the appendix as Figure A2:

"To conduct statistical tests, we used RStudio (version 1.0.153). We tested normality by using the Shapiro–Wilk test. Because our data does not follow a normal distribution, we used the Spearman rank correlation coefficient ($\rho$) to measure each relationship between DOC concentration and a further parameter (latitude, permafrost zone, ecoregion, ground ice content, deposit type, SOCC) for all lakes in our dataset. We used the Wilcoxon-Mann-Whitney test to determine the difference in means between two populations. To analyse the relationship of DOC and multiple parameters we performed a principal component analysis (PCA). Our dataset contains six samples from Qeqqata on Greenland (Osburn et al., 2017), collected in April with under-ice conditions. For the sake of comparability, these data have not been included in the statistical analysis."

We refer to the PCA in section 4.2 to 4.7:

"We found that lake DOC concentration was negatively correlated with geographic latitude of a lake ($\rho = -0.3$; $p < 0.05$; Table 4; Fig. A2). The DOC concentration of lakes in the southernmost study sites (Yukon Flats and Yukon-Charley Rivers National Preserve) showed a large range from 10.2 to 1,300 mg L-1, and 5.0 to 66.7 mg L-1, respectively (Table A3)."

"In our dataset, 43.7 % of the lakes were located in the boreal forest ecoregion, 42.6 % in the tundra region, and 13.7 % in a boreal-tundra transition zone. We found a significant relationship between lake DOC concentration and the lake surrounding ecoregion ($\rho$ = 0.31; $p < 0.05$; Table 4; Fig. A2), with significantly lower DOC concentrations in lakes of the tundra region ($p < 0.05$)."

"Median DOC concentrations were highest in lakes of the sporadic permafrost zone (17.3 mg L-1) and were negatively correlated with permafrost extent ($\rho$ = 0.37; $p < 0.05$; Fig. 3; Table 4; Fig. A2)."

"Our analysis shows a weak significant relationship of the lake surrounding deposit type and lake DOC concentration ($\rho$ = -0.2; $p < 0.05$; Table 4; Fig. A2)."

"We found a weakly positive relationship between ground ice content and lake DOC concentrations ($\rho$ = 0.05; $p < 0.05$; Table 4; Fig. A2)."

"We analysed the relationship between lake DOC concentrations and lake surrounding SOCC and found a weakly significant relationship for SOCC of the upper 100 cm ($\rho$ = 0.1; $p < 0.05$; Table 4; Fig. A2). The significance of the relationship was getting weaker for SOCC in the upper 300 cm ($\rho$ = 0.09; $p < 0.05$; Table 4, Fig. 4; Fig. A2)."

[Figure]

*Figure A2: Principal component analysis showing a variables factor map with a color gradient showing the contribution to the plane construction. The first two principal components explained 34.35 % (PC1) and 14.28 % (PC2) of the variance in the analysed parameters (DOC, latitude, permafrost zone, ecoregion, ground ice content, deposit type, SOCC). The scores of PC1 had*

*positive loadings with SOCC in all depths and negative loadings with deposit type, while PC2 scores had negative scores of latitude and positive scores of DOC, permafrost zone, ecoregion and ground ice content.*

Specific comments:

*Page 2, Line 10: better to specify which is the part of Siberia, where individual boreholes demonstrate significant positive temperature trend.*

We changed this accordingly as follows:

"More recently, permafrost warmed globally by an average of 0.29 °C +/- 0.12 °C over the 2007-2016 period due to higher air temperatures, with some of the strongest warming trends (about 0.9 °C per decade) measured in individual boreholes at the polar stations Marre Sale in northwest Siberia and Samoylov Island in northeast Siberia (Biskaborn et al., 2019)."

*Page 2, Line 25: is there direct chemical pathway from DOC to CH4? Anyway, this is not mineralization. If you mean that DOC is stored in sediments after flocculation, and then decomposed to CH4, please rephrase the sentence to make it clear.*

To clarify we rephrased the paragraph:

"In lakes, dissolved organic carbon (DOC) is one of the main C fractions (Tranvik et al., 2009). It is mobile and can be chemically labile (Vonk et al., 2013a, b). DOC in lakes can be produced in the lake itself (autochthonuous DOC) or in the catchment of the lake (allochthonuous DOC) (Sobek et al., 2007). The organic carbon (OC) content of terrestrial soils is the main source for allochthonuous DOC. DOC in lakes can be transferred to and stored in lake sediments due to flocculation (Tranvik et al., 2009). DOC can also be degraded by photo oxidation or microbial activity, resulting in the mineralization of OC to carbon dioxide ($CO_2$) and methane ($CH_4$) (Battin et al., 2008; Tranvik et al., 2009; Vonk et al., 2013a, b)."

*Page 2, Lines 26-29: "The mineralization of DOC in a lake is a major component of the global C cycle..." is too strong statement. "one third to one-half" – does this contribution include river C fluxes? Please rephrase to make understandable.*

To clarify our statement we rephrased the sentences as follows:

"DOC in lakes can be transferred to and stored in lake sediments due to flocculation (Tranvik et al., 2009). DOC can also be degraded by photo oxidation or microbial activity, resulting in the mineralization of OC to carbon dioxide ($CO_2$) and methane ($CH_4$) (Battin et al., 2008; Tranvik et

al., 2009; Vonk et al., 2013a, b). These processes are important components of the northern C cycle and affect greenhouse gas emissions from lakes."

"Vonk et al. (2015) suggested that the C flux from surface waters to the atmosphere and from land to ocean represents roughly one third to one-half of the net C exchange from land to the atmosphere in the Arctic."

*From which lacustrine layer DOC has been sampled in lakes? I guess, epilimnion. Please provide the info.*

Samples taken by the authors of this study were mostly taken from or near the water surface. Some of the DOC datasets we harvested from the literature do not provide information on the sampling depth in the meta-data or whether water bodies are stratified. A differentiation into epilimnion, metalimnion, hypolimnion would require CTD data, which presumably were not acquired in the vast majority of the synthesized data. Ultimately, many of our samples originate from shallow arctic lakes or ponds, which are known to be well mixed and rarely develop stratification.

We added a short information to section 3.1:

"Samples from the author team were taken from or near the water surface as well as the vast majority of the synthesized data. Although, some of the synthesized data do not provide the sampling depth we can assume that the majority of these arctic lakes and ponds are shallow and well mixed."

*Section 3.3: I would expect, that in order to isolate the influence of a single factor on lake DOC, you should compute correlation coefficient for a series of DOCs from all lakes, for which the other factors are fixed. Or you computed each correlation coefficient for the total number of lakes? Please precise.*

We rephrased section 3.3 find in the general comments and copied here:

"To conduct statistical tests, we used RStudio (version 1.0.153). We tested normality by using the Shapiro–Wilk test. Because our data does not follow a normal distribution, we used the Spearman rank correlation coefficient ($\rho$) to measure each relationship between DOC concentration and a further parameter (latitude, permafrost zone, ecoregion, ground ice content, deposit type, SOCC) for all lakes in our dataset. We used the Wilcoxon-Mann-Whitney test to determine the difference in means between two populations. To analyse the relationship of DOC and multiple parameters we performed a principal component analysis (PCA). Our dataset contains six samples from Qeqqata on Greenland (Osburn et al., 2017), collected in April with under-ice conditions. For the sake of comparability, these data have not been included in the statistical analysis."

*Table 4: not clear, how the rank correlation coefficient have been computed for qualitative predictors: permafrost region, ecoregion, ground ice content, deposit type. Please provide details.*

We revised the methodological description of our statistics find above and as follows:

"To conduct statistical tests, we used RStudio (version 1.0.153). We tested normality by using the Shapiro–Wilk test. Because our data does not follow a normal distribution, we used the Spearman rank correlation coefficient ($\rho$) to measure each relationship between DOC concentration and a further parameter (latitude, permafrost zone, ecoregion, ground ice content, deposit type, SOCC) for all lakes in our dataset. We used the Wilcoxon-Mann-Whitney test to determine the difference in means between two populations. To analyse the relationship of DOC and multiple parameters we performed a principal component analysis (PCA). Our dataset contains six samples from Qeqqata on Greenland (Osburn et al., 2017), collected in April with under-ice conditions. For the sake of comparability, these data have not been included in the statistical analysis."

*What is the reason for ultrahigh DOC concentrations in some Alaskan lakes?*

At this point, we did not find the ultimate cause for the very high DOC concentrations in some of the lakes in the Yukon Flats in Interior Alaska. We discussed possible causes in the sections of our discussion. In 5.1 we described the connection between DOC concentration and ecoregion, also referring to the high concentrations in lakes in the Yukon Flats:

"We particularly found that lakes in the boreal forest region have higher DOC concentrations compared to tundra region lakes (Fig. 3). Soils of boreal forests are rich in organic material and microbial degradation is low (Sobek et al. 2007). In areas of boreal forest, the frost-free period is extended and the surface water can be in contact with soil C for a longer time resulting in higher DOC concentrations in boreal lakes. Previous studies confirm that vegetation is an important driver for DOC in permafrost catchments (Harms et al., 2016; Coch et al., 2019). Coch et al. (2019) found higher DOC concentrations in moss and plant rich Low Arctic catchments on Herschel Island in Northwest Canada compared to a High Arctic catchments at Cape Bounty, Northeast Canada. In our database we found high lake DOC concentrations in the boreal forest regions of Interior Alaska which are dominated by white and black spruce (Halm & Griffith, 2014)."

To clarify we rephrased the last sentence of the upper paragraph:

"This relationship may explain high lake DOC concentrations we found in the Yukon Flats in Interior Alaska, a study area in the boreal forest and dominated by white and black spruce (Halm & Griffith, 2014)."

Additionally, we discussed the high DOC concentrations in the Yukon Flats in section 5.2:

"While we showed that lake DOC concentration is influenced by permafrost extent and type of ecoregion they do not explain all of the variability in the dataset. Additional factors are regulating DOC. For example, air temperature, precipitation and solar radiance have an influence on surface water DOC concentration (Cole et al., 2002; Molot et al., 2005; Anderson & Stedmon, 2007). Anderson & Stedmon (2007) analysed lakes in low Arctic Greenland and found highest lake DOC concentrations in areas of low precipitation and low discharge. In those areas, evaporation is high leading to higher DOC concentrations. For our database, the role of evaporation may also explain the relatively high DOC concentrations of lakes in the Yukon basin. Here, the lakes are less hydrologically connected and the region is very arid, allowing evaporation-driven concentration of DOC (Johnston et al., 2020)."

Here, we also rephrased the penultimate sentences of the upper paragraph to be more precise:

"For our database, the role of evaporation may also explain the high DOC concentrations of lakes in the Yukon Flats in Interior Alaska. Here, the lakes are less hydrologically connected and the region is very arid, allowing evaporation-driven concentration of DOC (Johnston et al., 2020). "

Finally, we discussed the high DOC concentrations in Yukon Flats lakes in section 5.3 and rephrased to give more information:

"However, it is known for example that less allochthonous DOC is transported to a hydrologically isolated lake than to a connected lake (Bogard et al., 2019). In arid regions with rather isolated lakes, such as in the Yukon Flats in Interior Alaska, evapoconcentration of DOC plays an important role (Johnston et al., 2020). Water bodies with highest DOC concentrations in the Yukon Flats have a water depth less than 1 m. Studies in West-Siberia showed, that ponds receive the highest impact of allochthonous input due to the high ratio of lake drainage area vs. small water volumes. This results in short water residence time leading to highest concentrations of DOC (Shirokova et al., 2013; Manasypov et al., 2014, 2015). In addition to allochthonous DOC, autochthonous DOC, including phytoplankton productivity as well as heterotrophic bacterioplankton respiration processes (Chupakov et al., 2017), is influencing the DOC concentration, especially in lakes with low connectivity. For lakes in the Yukon River Basin, Bogard et al. (2019) described a minor importance of allochthonous DOC in lakes and highlighted the carbon fixation from atmospheric $CO_2$."

*Page 16, Lines 30-31: an unfinished sentence*

Thank you very much. We changed this accordingly:

"This might be attributed to the mobilization of old labile yedoma carbon by thermo-erosion along rapidly expanding lake shores and by active thermokarst processes (Strauss et al., 2017)."

*Page 17, Line 22: I guess you meant "occupied by wetlands"*

We changed it accordingly:

"For example, Xenopoulos et al. (2003) analysed catchment characteristics of lakes and found that lake perimeter and the proportion of the watershed occupied by wetlands are strongest predictors for DOC in lakes of temperate forests. On a global scale, lake elevation and the proportion of wetlands in a watershed are strongest predictors for lake DOC."